# Improving Task-Specific Generalization in Few-Shot Learning via Adaptive Vicinal Risk Minimization

**Long-Kai Huang**
Tencent AI Lab
hlongkai@gmail.com

**Ying Wei**[*]
City University of Hong Kong
yingwei@cityu.edu.hk

## Abstract

Recent years have witnessed the rapid development of meta-learning in improving the meta generalization over tasks in few-shot learning. However, the task-specific level generalization is overlooked in most algorithms. For a novel few-shot learning task where the empirical distribution likely deviates from the true distribution, the model obtained via minimizing the empirical loss can hardly generalize to unseen data. A viable solution to improving the generalization comes as a more accurate approximation of the true distribution; that is, admitting a Gaussian-like vicinal distribution for each of the limited training samples. Thereupon we derive the resulting vicinal loss function over vicinities of all training samples and minimize it instead of the conventional empirical loss over training samples only, favorably free from the exhaustive sampling of all vicinal samples. It remains challenging to obtain the statistical parameters of the vicinal distribution for each sample. To tackle this challenge, we further propose to estimate the statistical parameters as the weighted mean and variance of a set of unlabeled data it passed by a random walk starting from training samples. To verify the performance of the proposed method, we conduct experiments on three standard few-shot learning benchmarks and consolidate the superiority of the proposed method over state-of-the-art few-shot learning baselines.

## 1 Introduction

In recent years, deep learning algorithms have made tremendous progress and have been widely applied in vision [16], language [37] and speech [10] areas. Despite the prevalence, deep learning algorithms rely heavily on a sufficient amount of labeled data and often fail when only a limited number of labeled samples are available. Considering the high cost of extensive data collection and annotation, it has attracted increasing attention to developing algorithms learning from a limited number of training samples, which has been widely known as few-shot learning [39, 5].

The recent focus of few-shot learning is built on the meta-learning framework [9, 39, 33]. The goal of meta-learning is to train a model that can be quickly adapted to a new task with a small number of labeled samples by leveraging the knowledge from previous related tasks. Typically, a meta-learning model consists of the meta-training stage where a well-generalized feature extractor is learned from meta-training tasks and the meta-testing stage during which a classifier is built on top of the feature extractor for novel tasks. While the majority of meta-learning works revolved around developing better meta-training strategies to reduce the meta-generalization error over all tasks, how to fully leverage the meta-trained feature extractor to build a robust classifier and improve a task-specific generalization in the meta-testing stage remains less explored.

---

[*]Correspondence to Ying Wei.

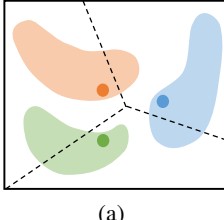 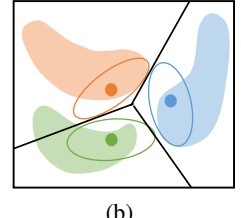 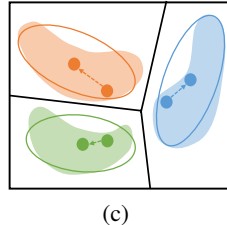

| (a) | (b) | (c) |

Figure 1: (a) The classifier overfits to few-shot examples that inadequately represent the true distribution. (b) The vicinal distribution without adjusting the means slightly improves the classifier. (c) The vicinal distribution with updated means contributes to a well-generalizable classifier.

The challenge to building such a classifier lies in the significant deviation of the empirical distribution (i.e. the distribution formed by the training data) from the population distribution (i.e. the true distribution). As illustrated in Figure 1, a model well fitting the training data (support set) may generalize poorly on the testing data (query set) from the true distribution due to the bias in the empirical distribution. One viable solution is to approximate the true distribution more accurately – by viewing the empirical distribution as the assembling of delta functions located on each training sample, Chapelle et al. [4] proposed the Vicinal Risk Minimization loss function where the delta functions are replaced by some density estimate in the vicinity of each training data. In the era of deep learning, the VRM learning principle is implemented as Mixup [47, 38], which is simple yet effective given a sufficiently large number of training data. However, as we have verified in Table 5, the performance improvement by applying Mixup on the support set is limited in few-shot learning where the vicinal distributions assembled by Mixup is still significantly biased. To improve the generalization of a few-shot classifier during meta-testing, we are motivated to revisit the VRM principle to 1) generate adaptive vicinal distribution for each sample in support set and 2) minimize the expected vicinal loss over the vicinal distributions. However, both are challenging.

Without loss of generality, we assume that the vicinity of each training point admits a Gaussian distribution. In [4], the means of the vicinal distributions are located on the original training points. As illustrated in Figure 1(b), however, setting the mean in this way is not sufficient to approximate the true distribution, as an original training point likely deviates from the mean of the ground-truth distribution of its corresponding class. In fact, it is difficult to estimate the mean and variance based only on the labeled training samples. In this paper, we assume that unlabeled samples are also available, and leverage these unlabeled samples to estimate the mean and variance of the vicinal distribution for each labeled sample. Concretely, inspired by [48], we first construct a transition probability matrix based on pairwise distances between features of all samples, and secondly perform the lazy random walk among unlabeled samples starting from each labeled sample. After walking a certain number of steps, we obtain the visit probabilities of all the samples and infer the vicinal distribution whose mean and variance are the weighted mean and variance of all samples visited.

All that remains is the loss function over the vicinal distributions of all training samples. An intuitive way to do this is to exploit the Monte Carlo method, which repeats random sampling to explicitly generate data points from each vicinal distribution and minimizes the loss on all the sampled data points. Unfortunately, it requires drawing a large amount of samples to minimize the Monte Carlo approximation error, which significantly increases the computation cost. Instead, we follow [4] to derive a closed form of the expected vicinal loss over the vicinal distributions. In this paper, we focus on building an optimization-based classifier in optimization-based meta-learning algorithms [9, 19], which has been proved with much more adaptability and generalization capacity in novel meta-testing tasks than a nearest-neighbour classifier in metric-based meta-learning algorithms [39, 33, 34, 24]. Particularly, we consider both the cross-entropy loss and the SVM loss functions which are used as classifier in the state-of-the-art meta-learning algorithms [43, 19]. For the vicinal version of cross-entropy loss, it cannot be computed in closed form. We instead derive its quadratic approximation. For SVM, we derive its closed-form vicinal loss.

In summary, our contributions are threefold:

- We revisit vicinal risk minimization (VRM) in deep few-shot learning and derive the resulting expected vicinal loss functions for both the cross-entropy loss and SVM.

- We propose to harness unlabeled data and a lazy random walk algorithm to generate adaptive statistical parameters for the vicinal distribution of each training sample.
- We have validated the superiority of the proposed algorithm with extensive experiments on three benchmarks, i.e., miniImageNet, CUB, CIFAR-FS.

## 2 Related Works

### 2.1 Few-Shot Learning and Meta-Learning

Few-shot learning aims to recognize novel classes with insufficient labeled instances. The recent focus of few-shot learning algorithms has been on the meta-learning framework, which trains the model with multiple episodes to enable quick learning and applies the model to improve learning in novel tasks. The meta-learning methods fall into two major categories: metric-based methods and optimization-based methods. Metric-based methods [15, 39, 33, 34, 23, 45] learn a metric to compare two samples' feature embeddings encoded by the feature extractor and classify a sample in the query set based on its distances to the samples in the support set. Optimization-based approaches [9, 27, 31, 35, 41] learn a good initialization such that it can be fast adapted to novel tasks based on a few instances.

To tackle the data deficiency issue in few-shot learning, researchers have explored the idea of exploiting unlabeled data and developed transductive learning or semi-supervised learning techniques for few-shot learning, either by leveraging nearest neighbor graph for propagating labels [42, 14, 21] or by leveraging predicted soft or hard labels on unlabeled samples to update the class prototypes or metric [29, 12, 17, 26, 45]. Our method is not built on a metric-based method. We make use of the unlabeled data to derive the vicinal distribution for each support sample and optimize the consequential vicinal loss to learn a classifier. Our approach shares a similar spirit of [43]. Yang et al. [43] proposed to calibrate the distribution of the support samples by transferring statistics from the base classes and trained the classifier on a set of samples sampled from the calibrated distributions. Unlike [43], our method obtains more accurate vicinal distributions by making use of the unlabeled data from the same task. Besides, instead of sampling extensive data from the distributions, our model directly optimizes the expected vicinal loss over the distributions, resulting in more efficient training and higher performance.

### 2.2 Vicinal Risk Minimization

The Vicinal Risk Minimization (VRM) principle is introduced in [4, 36]. It assumes each training sample admits a vicinal distribution and optimizes the expected loss over the vicinal distributions. In [4], only binary VRM classifiers are considered. Our method considers multi-class classifiers and the cross-entropy loss, which is more challenging to derive the expected vicinal loss. Besides, Chapelle et al. [4] consider Gaussian vicinities but only estimate the variances, keeping the mean the same as the original training sample. Our approach further estimates the mean with the unlabeled data.

In the era of deep learning, the most popular technique under the VRM principle is Mixup [47, 38]. Mixup constructs the vicinal distributions as the convex combination of the training data, which can be understood as a data augmentation technique that encourages the model to behave linearly in-between training examples. It has been demonstrated that Mixup is simple yet efficient and improves the generalization of deep models given sufficiently many training samples. Mixup has been introduced on the meta-learning algorithm in [44, 6], but is applied to the query set, focusing on improving the meta generalization. As the experiment results in [44] (Table 6) and in Table 5 shows, applying Mixup on the support set could even hurt the performance. Our method instead improves the performance by producing an accurate estimate of the vicinal distribution for the small support set using unlabeled data.

## 3 Adaptive Vicinal Few-Shot Learning

In this section, we will introduce the proposed algorithm, **AD**aptive **V**icinal Few-Shot Learning (ADV). We will first introduce the VRM, then derive the expected vicinal loss for a linear classifier train with cross-entropy loss and for SVM, respectively, and finally propose the algorithm based on lazy random walks to obtain the adaptive vicinal distribution for each labeled support sample. We summarize the proposed method in Algorithm 1. Note that our method focuses on improving the

task-specific generalization in meta-testing set and is compatible with feature extractor trained by any meta-learning algorithm.

## 3.1 Vicinal Risk Minimization

Given data $x, y$ following the distribution $P(x, y)$, the loss function $\mathcal{L}$, we would like to optimize a model $f_\theta(x)$ by minimizing the expected risk

$$\mathcal{L}(\theta) = \int \ell(f_\theta(x), y) \, \mathrm{d}P(x, y).$$

Here, $x$ is the input data, $y$ is its label and $\theta$ is the model parameters. In practice, $P(x, y)$ is unknown and a set of training data $\{x_i, y_i\}_{i=1}^N$ is available, where $N$ is the size of training data. To learn a model, one can minimize the empirical risk, which is equivalent to minimizing the expected loss with respect to an empirical density estimate $\mathrm{d}P_\delta(x, y) = \frac{1}{N} \sum_i^N \delta_{x_i}(x) \delta_{y_i}(y)$, where $\delta_a(\cdot)$ is a Dirac delta function centered at $a$.

Given a limited number of training sample, the empirical density estimate $\mathrm{d}P_\delta(x, y)$ could bias from the true density $\mathrm{d}P(x, y)$. For a more accurate density estimate, Chapelle et al. [4] proposed to replace the delta functions $\delta_{x_i}(x)$ by some estimate of density in the vicinity of $x_i$. Given a vicinal distribution of $x_i$ as $P_{x_i}(x)$, the improved density estimate is $P_\nu(x, y) = \frac{1}{N} \sum_{i=1}^N P_{x_i}(x) \delta_{y_i}(y)$. And the corresponding vicinal risk is defined as

$$\mathcal{L}_\nu(\theta) = \int \ell(f_\theta(x), y) \, \mathrm{d}P_\nu(x, y) = \frac{1}{N} \sum_{i=1}^N \int \ell(f_\theta(x), y_i) \, \mathrm{d}P_{x_i}(x).$$

## 3.2 Vicinal Loss

In this paper, we assume the feature extractor is trained and fixed for novel tasks. Each task is $C$-way $K$-shot, consisting of $N = CK$ data support samples $\{(x_i, y_i)\}_{i=1}^N$. To perform VRM, we assume the feature $z_i$ of each sample $x_i$ in the support set admits a Gaussian vicinal distribution $\mathcal{N}(\mu_i, \Sigma_i)$ such that a sample $\tilde{z}$ is drawn from the vicinal distribution with probability $P_{z_i}(\tilde{z}) = \det(2\pi\Sigma_i)^{-\frac{1}{2}} \exp(-\frac{1}{2}(\tilde{z} - \mu_i)^\top \Sigma_i^{-1}(\tilde{z} - \mu_i))$, where $\mu_i$ and $\Sigma_i$ is the mean and variance of the vicinal distribution of $z_i$.

### 3.2.1 Vicinal Cross-Entropy Loss

We first focus on linear classifier with Cross-Entropy loss. A linear classifier $f_\theta(z) = (W^\top z + b)$ maps a feature $z \in \mathbb{R}^d$ to its logits, where $W \in \mathbb{R}^{d \times C}$ and $b \in \mathbb{R}^C$ is the parameter of the classifier.

The cross-entropy loss is defined as $\ell(h, y) = -e_y^\top \log(\mathrm{softmax}(f_\theta(z)))$, where $y \in 1, 2, ..., C$ is the label; $e_y$ is a one-hot vector with the $y$-th element being 1. The expected loss of cross entropy over Gaussian vicinal distribution of $x$ is defined as

$$\ell_\nu(f_\theta(z), y) = \int -e_y^\top \log(\mathrm{softmax}(f_\theta(\tilde{z}))) \, \mathrm{d}P_z(\tilde{z}) = \int \ell(f_\theta(\tilde{z}), y) \, \mathrm{d}P_z(\tilde{z}) = \mathbb{E}_{P_z(\tilde{z})}[\ell(f_\theta(\tilde{z}), y)]. \quad (1)$$

Due to the softmax function, the expected loss (1) cannot be computed in closed form. Instead, we utilize Taylor expansion to derive an approximation of Eqn.(1). By taking the second-order Taylor expansion of $\ell(f_\theta(\tilde{z}), y)$ around the mean of the vicinal distribution $z_i$, i.e. $\mu$, we obtain

$$\ell(f_\theta(\tilde{z}), y) \approx \ell(f_\theta(\mu), y) + G(\tilde{z} - \mu) + \frac{1}{2}(\tilde{z} - \mu)^\top H(\tilde{z} - \mu), \quad (2)$$

where $G$ is the gradient and $H$ is the Hessian w.r.t. input $\mu$. Note that $H = W(\mathrm{diag}(h) - hh^\top)W^\top$ where $h = \mathrm{softmax}(f_\theta(\mu))$ and $\mathrm{diag}(h)$ is a diagonal matrix whose diagonal elements are equal to $h$ [32]. By taking the expectation over $\tilde{z}$, we have

$$\mathbb{E}_{P_z(\tilde{z})}[G(\tilde{z} - \mu)] = 0, \quad (3)$$

$$\mathbb{E}_{P_z(\tilde{z})}[(\tilde{z} - \mu)^\top H(\tilde{z} - \mu)] = \sum_{k=1}^C (h_k W_k^\top \Sigma W_k) - \sum_{j=1}^C \sum_{k=1}^C h_j h_k W_j^\top \Sigma W_k, \quad (4)$$

---

**Algorithm 1** Adaptive Vicinal Few-Shot Learning (ADV)

---

1: **Input:** Support data $\{z_i, y_i\}_{i=1}^N$, unlabeled data $\{u_j\}_{j=1}^{N_u}$, lazy stay parameter $\beta$, number of step in lazy random walk $T$
2: Compute the distance matrix $D^{SU}$ and $D^{UU}$ by (10) and (11)
3: **for** all $z_i$ **do**
4:     Compute the transition probabilities matrix $P_i$ by (12) and (13)
5:     Compute the $T$-step visit probability $v_i$ by (14)
6:     Compute $\mu_i$ and $\Sigma_i$ as the weighted mean of variance of $z_i$ and $\{u_j\}_{j=1}^{N_u}$ with the weights being $v_i$
7: **end for**
8: Compute the objective as the vicinal cross-entropy loss in (5) or as the Vicinal SVM in (9)
9: Optimize the objective until convergence

---

where $\Sigma$ is the variance of $\tilde{z}$, $h_k$ is the $k$-th elements of $h$, $W_k$ is the $k$-th column of $W$. We define $R_\theta(\Sigma)$ as the RHS of Eqn. (4). Plugging (2), (3) and (4) into (1), we obtain the approximated expected vicinal cross-entropy loss for the linear classifier as

$$\mathcal{L}_\nu(\theta) = \frac{1}{N} \sum_{i=1}^N \left[ \ell(f_\theta(\mu_i), y_i) + \frac{1}{2} R_\theta(\Sigma_i) \right]. \tag{5}$$

### 3.2.2 Vicinal SVM

SVM is used as classifier for meta-learning algorithms in [19, 43]. We follow [19] to use the Crammer and Singer formulation of multi-class SVM [7]. In [19, 7], the dual formulation of the objective for a linear multi-class SVM is obtained as:

$$\max_{\alpha^k} \left[ -\frac{1}{2} \sum_{k=1}^C \sum_{i,j=1}^N (\alpha_i^k \alpha_j^k) z_i^\top z_j + \sum_{i=1}^N \alpha_i^{y_i} \right] \tag{6}$$

$$s.t. \ \alpha_i^{y_i} \leq \lambda; \ \alpha_i^k \leq 0 \ \forall k \neq y_i; \ \sum_{k=1}^C \alpha_i^k = 0 \ \forall i,$$

where $\alpha_i^k$ is the dual variable and $\lambda$ is a regularization parameter. The linear classifier for the $k$-th class is $f_\theta^k(z) = w_k^\top z$, where $w_k = \sum_{i=1}^N \alpha_i^k z_i$. Note that here $z$ is an unseen data. We assume only labeled data admit vicinal distributions while unseen data does not admit vicinal distributions. If each $z_i$ admits a vicinal distribution $\mathcal{N}(\mu_i, \Sigma_i)$, the vicinal expectation for the term $z_i^\top z_j$ in (6) becomes

$$\mathbb{E}_{\tilde{z}_i \sim \mathcal{N}(\mu_i, \Sigma_i)} \mathbb{E}_{\tilde{z}_j \sim \mathcal{N}(\mu_j, \Sigma_j)} [\tilde{z}_i^\top \tilde{z}_j] = \mu_i^\top \mu_j,$$

and the classifier becomes

$$\mathbb{E}_{\tilde{z}_i \sim \mathcal{N}(\mu_i, \Sigma_i)} [f_\theta^k(z)] = \left( \sum_{i=1}^N \alpha_i^k \mu_i \right)^\top z.$$

If $\mu_i = z_i$, the objective of the vicinal SVM reduces to the original SVM objective in Eqn. (6). Even when $\mu_i \neq z_i$, the objective of the vicinal SVM still does not take the variance of the vicinal distribution into consideration and may result in suboptimal performance. To exploit the variance information to improve the vicinal SVM classifier, we consider a nonlinear classifier, i.e. the kernel-based SVM.

Multi-class kernel-based SVM uses a kernel $\mathcal{K}(\cdot, \cdot)$. It admits a dual formulation similar to (6), but replaces the inner product of $z_i$ and $z_j$ in (6) by $\mathcal{K}(z_i, z_j)$. And the classifier becomes $f_\theta^k(z) = \sum_i \alpha_i^k \mathcal{K}(z_i, z)$. In this paper, we consider the widely-used RBF kernel $\mathcal{K}(z, z') = \exp(-\frac{||z-z'||^2}{2\sigma^2})$, where $\sigma$ is a hyper-parameter.

In vicinal multi-class kernel-based SVM, we define $\mathcal{M}(z_i, z_j) = \mathbb{E}_{\tilde{z}_i \sim \mathcal{N}(\mu_i, \Sigma_i)} \mathbb{E}_{\tilde{z}_j \sim \mathcal{N}(\mu_j, \Sigma_j)} \mathcal{K}(z_i, z_j)$ and $\mathcal{Q}(z_i, z) = \mathbb{E}_{\tilde{z}_i \sim \mathcal{N}(\mu_i, \Sigma_i)} \mathcal{K}(z_i, z)$ and obtain their computation as

$$\mathcal{Q}(z_i, z) = \frac{|A(\sigma^2)|^{\frac{1}{2}}}{|A(\sigma^2) + \Sigma_i|^{-\frac{1}{2}}} \exp\left( -\frac{1}{2} ||z - \mu_i||_{M_i}^2 \right), \tag{7}$$

$$\mathcal{M}(z_i, z_j) = \frac{|A(\sigma^2)|^{\frac{1}{2}}}{|A(\sigma^2) + \Sigma_i + \Sigma_j|^{-\frac{1}{2}}} \exp\left( -\frac{1}{2} ||\mu_i - \mu_j||_{M_{ij}}^2 \right). \tag{8}$$

The detailed derivation is available in Appendix A. Here, $A(\sigma^2) \in \mathbb{R}^{d \times d}$ is a diagonal matrix with diagonal elements being $\sigma^2$; $M_i = (A(\sigma^2) + \Sigma_i)^{-1}$; $M_{ij} = (A(\sigma^2) + \Sigma_i + \Sigma_j)^{-1}$; the operator $\|z\|_M^2 = z^\top M z$. Note that for computation simplification, in vicinal SVM, we assume the variance $\Sigma_i$ is diagonal and set its non-diagonal elements to zeros.

Now we achieve the vicinal objective for multi-class kernel-based SVM as:

$$\max_{\alpha^k} \left[ -\frac{1}{2} \sum_{k=1}^{C} \sum_{i,j=1}^{N} (\alpha_i^k \alpha_j^k) \mathcal{M}(z_i, z_j) + \sum_{i=1}^{n} \alpha_i^{y_i} \right] \tag{9}$$

$$s.t. \ \alpha_i^{y_i} \leq \lambda; \ \alpha_i^k \leq 0 \ \forall k \neq y_i; \ \sum_k \alpha_i^k = 0 \ \forall i,$$

with the classifier being $f_\theta^k(z) = \sum_i \alpha_i^k \mathcal{Q}(z_i, z)$. The objective in (9) is a quadratic program (QP) over dual variables $\{\alpha_i^k\}$ and can be solved by using a differentiable GPU-based QP solver [1].

### 3.3 Lazy Random Walk for an Adaptive Vicinal Distribution

As illustrated in Figure 1, the performance of the model learned by VRM relies heavily on how close the vicinal distribution is to the true distribution. However, it is almost impossible to estimate accurate proximate distributions from a small support set. Therefore, we propose to exploit extra data, i.e., the unlabeled data from the same task, to reduce the estimation error of the statistical parameters for the vicinal distribution and, thus, improve the performance of the model.

We assume that a well-trained feature extractor on the meta-training tasks can preserve the similarity between the data in the feature space. Data from the same class in the novel task would have similar features with a high probability. Then, we apply the lazy random walk algorithm [48] to estimate the probability that the unlabeled data will be visited starting from a support sample, and compute the statistical parameters as the weighted mean and variance of the samples visited by the lazy random walk.

To perform lazy random walk, we first compute the distance $D^{SU}$ between the support sample $z_i$ and the unlabeled data $\{u_j\}_{j=1}^{N_u}$ and compute the distance $D_{UU}$ between the unlabeled data as:

$$D_{ij}^{SU} = \|z_i - u_j\|^2, \quad \text{and} \tag{10}$$

$$D_{ij}^{UU} = \begin{cases} \|u_i - u_j\|^2 & i \neq j \\ \infty & i = j \end{cases} \tag{11}$$

Secondly, we transform these distances into transition probabilities by softmaxing the distance matrix over columns with a temperature $\tau$:

$$P_{ij}^{SU} = exp\left(-\frac{1}{\tau} D_{ij}^{SU}\right) / \sum_{j'} exp\left(-\frac{1}{\tau} D_{ij'}^{SU}\right),$$

$$P_{ij}^{UU} = exp\left(-\frac{1}{\tau} D_{ij}^{UU}\right) / \sum_{j'} exp\left(-\frac{1}{\tau} D_{ij'}^{UU}\right). \tag{12}$$

Note that we set $D_{jj}^{UU} = \infty$ such that $P_{jj}^{UU} = 0$.

Thirdly, we construct the transition probabilities matrix for $z_i$ in lazy random walk by combining $P_{i:}^{SU}$ and $P^{UU}$ and setting the lazy stay probability $\beta \in (0, 1)$ as:

$$\mathcal{P}_i = \begin{bmatrix} \beta, & (1-\beta)P_{i:}^{SU} \\ 0_{N_u \times 1}, & \beta I_{N_u} + (1-\beta)P^{UU} \end{bmatrix}, \tag{13}$$

where $P_{i:}^{SU}$ is the $i$-th row of $P^{SU}$, $0_{n_c \times 1}$ is a all zero vector, and $I_{N_u}$ is an identity matrix.

We then compute the visit probability $v_i^t$ at the $t$-th step for $z_i$:

$$v_i^t = e_1^\top \mathcal{P}_i^t, \tag{14}$$

where $e_1 \in \mathbb{R}^{(N_u+1) \times 1}$ is a one-hot vector with the first element being 1. For computational efficiency, we keep only the top $m$ nearest neighbors of a data visitable and set $D_{ij}^{UU}$ ($D_{ij}^{SU}$) to zero if $u_j$ is not in the top $m$ nearest neighbors of $u_i$ ($z_i$). Finally, we obtain $\mu_i$ and $\Sigma_i$ as the weighted

Table 1: Comparison with baselines on 5-way 1-shot and 5-way 5-shot classification accuracy (%) on miniImageNet, CUB and CIFAR-FS with 95% confidence intervals. For the backbone, CONV denotes a network consisting of 4 convolution layer; ResNet denotes ResNet-12; WRN denotes WideResNet-28. The best two accuracies are highlighted in bold.

| Methods | Backbone | miniImageNet 5-way | | CUB 5-way | | CIFAR-FS 5-way | |
|---|---|---|---|---|---|---|---|
| | | 1-shot | 5-shot | 1-shot | 5-shot | 1-shot | 5-shot |
| MAML [9] | CONV | $48.70 \pm 1.84$ | $63.11 \pm 0.92$ | $50.45 \pm 0.97$ | $59.60 \pm 0.84$ | $58.9 \pm 1.9$ | $71.5 \pm 1.0$ |
| MetaSGD [20] | CONV | $50.47 \pm 1.87$ | $64.03 \pm 0.94$ | $53.34 \pm 0.97$ | $67.59 \pm 0.82$ | - | - |
| MatchningNet [39] | CONV | $43.56 \pm 0.84$ | $55.31 \pm 0.73$ | $56.53 \pm 0.99$ | $63.54 \pm 0.85$ | - | - |
| ProtoNet [33] | CONV | $49.42 \pm 0.78$ | $68.20 \pm 0.66$ | $58.43$ | $75.22$ | $55.5 \pm 0.7$ | $72.0 \pm 0.6$ |
| Relation Net [34] | CONV | $50.44 \pm 0.82$ | $65.32 \pm 0.70$ | $62.45$ | $76.11$ | $55.0 \pm 1.0$ | $69.3 \pm 0.8$ |
| R2D2 [2] | CONV | $51.2 \pm 0.6$ | $68.8 \pm 0.1$ | - | - | $65.3 \pm 0.2$ | $79.4 \pm 0.1$ |
| SNAIL [23] | ResNet | $55.71 \pm 0.99$ | $68.88 \pm 0.92$ | - | - | - | - |
| TADAM [24] | ResNet | $58.50 \pm 0.30$ | $76.70 \pm 0.30$ | - | - | - | - |
| MetaOptNet-SVM [19] | ResNet | $62.64 \pm 0.61$ | $78.63 \pm 0.46$ | - | - | $72.0 \pm 0.7$ | $84.2 \pm 0.5$ |
| Meta-Fun [41] | ResNet | $62.12 \pm 0.30$ | $78.20 \pm 0.16$ | - | - | - | - |
| LEO [31] | WRN | $61.76 \pm 0.08$ | $77.59 \pm 0.12$ | - | - | - | - |
| S2M2 [22] | WRN | $64.93 \pm 0.48$ | $83.18 \pm 0.72$ | $80.68 \pm 0.81$ | $90.85 \pm 0.44$ | $74.81 \pm 0.19$ | $\mathbf{87.47 \pm 0.13}$ |
| FreeLunch [43] | WRN | $68.57 \pm 0.55$ | $82.88 \pm 0.42$ | $79.56 \pm 0.87$ | $90.67 \pm 0.35$ | - | - |
| Transductive Methods | | | | | | | |
| TPN [21] | CONV | $55.51 \pm 0.86$ | $69.86 \pm 0.65$ | - | - | - | - |
| FEAT [45] | CONV | $57.04 \pm 0.20$ | $72.89 \pm 0.16$ | - | - | - | - |
| TEAM [26] | CONV | $56.57$ | $72.04$ | $75.71$ | $86.04$ | - | - |
| TPN [21] | ResNet | $59.46$ | $75.65$ | - | - | - | - |
| TEAM [26] | ResNet | $60.07$ | $75.90$ | $80.16$ | $87.17$ | - | - |
| SIB [13] | WRN | $70.0 \pm 0.6$ | $79.2 \pm 0.4$ | - | - | $80.0 \pm 0.6$ | $85.3 \pm 0.4$ |
| ADV-SVM | WRN | $\mathbf{72.18 \pm 0.48}$ | $\mathbf{84.38 \pm 0.30}$ | $\mathbf{83.25 \pm 0.44}$ | $\mathbf{91.23 \pm 0.23}$ | $\mathbf{81.12 \pm 0.46}$ | $87.41 \pm 0.33$ |
| ADV-CE | WRN | $\mathbf{74.63 \pm 0.48}$ | $\mathbf{86.12 \pm 0.29}$ | $\mathbf{84.20 \pm 0.43}$ | $\mathbf{91.54 \pm 0.22}$ | $\mathbf{82.02 \pm 0.48}$ | $\mathbf{88.25 \pm 0.31}$ |

mean and variance of $z_i$ and $\{u_j\}_{j=1}^{N_u}$, with the weight being $v_i^t$, averaged over $T + 1$ steps, where in step 0 it visits only $z_i$. That is:

$$\mu_i = \frac{1}{T+1} \left( z_i + \sum_{t=1}^{T} \sum_{j=1}^{N_u} v_{ij}^t u_j \right)$$

$$\Sigma_i = \frac{1}{T+1} \left( (z_i - \mu_i)(z_i - \mu_i)^\top + \sum_{t=1}^{T} \sum_{j=1}^{N_u} v_{ij}^t (u_j - \mu_i)(u_j - \mu_i)^\top \right) .$$

## 4 Experiments

In this section, we verify the effectiveness of the proposed approach in improving the generalization of the few-shot learning. We first introduce the implementation details. Then we present the results on the benchmarks, followed by a detailed study of the proposed approach.

We use three benchmarks for performance evaluation: *mini*ImageNet [39], CUB [40] and CIFAR-FS [2]. For all datasets, we construct 5-way 1-shot and 5-way 5-shot tasks by randomly sampling 5 classes. For performance evaluation, we construct $10,000$ meta-testing tasks and report the average classification accuracy with a 95% confidence interval.

**Implementation details**: we focus on meta-testing phase and directly use the WideResNet-28-10 (WRN) [46] feature extractors pre-trained following the previous work [22]. We use ADV-CE to denote the application of ADV in the linear classifier with vicinal cross-entropy loss and ADV-SVM to denote the application of ADV in SVM. For ADV-CE, we initialize the weights of the classifier by class prototypes and optimize the vicinal loss in (5) by gradient descent for 100 steps. The learning rate is determined by performing a grid search from $0.001$ to $1$ on the tasks constructed by the meta-validation set. For ADV-SVM, we solve the QP in (9) by using a differentiable GPU-based QP solver [1]. The regularization parameter $\lambda$ is set to $0.1$ and the parameter $\sigma$ for RBF kernel is obtained via grid search from $0.1$ to $10$. In the lazy random walk algorithm, the number of steps $T$, the lazy stay probability $\beta$ and the temperature $\tau$ are obtained via grid search in $\{1, 2, 3, 4, 5\}$, $\{0.1, 0.2, 0.5\}$, $\{0.01, 0.1, 1, 10\}$, respectively. Notably, ADV accesses the whole query set as unlabeled data to perform lazy random walk. Thus, ADV is a transductive method. The number of query samples per class is 15.

Table 2: Comparision with baselines using the same fixed pre-trained WRN feature extractor on 5-way 1-shot and 5-way 5-shot classification accuracy (%) on miniImageNet with 95% confidence interval.

| Method | 1-shot | 5-shot |
|---|---|---|
| TIM [3] | $74.36 \pm 0.26$ | $\textbf{84.58} \pm \textbf{0.15}$ |
| LaplacianShot [49] | $69.10 \pm 0.23$ | $80.86 \pm 0.15$ |
| OM [25] + TIM | $68.67 \pm 0.23$ | $83.42 \pm 0.15$ |
| ADV-CE | $\textbf{74.92} \pm \textbf{0.25}$ | $84.53 \pm 0.16$ |

Table 3: Comparision with baselines on 5-way 1-shot and 5-way 5-shot classification accuracy (%) on miniImageNet with 95% confidence interval.

| Method | 1-shot | 5-shot |
|---|---|---|
| TIM [3] | 77.8 | 87.4 |
| LaplacianShot [49] | $74.86 \pm 0.19$ | $84.13 \pm 0.14$ |
| [8] + TIM | 80.04 | 87.64 |
| [18] | 76.84 | 84.36 |
| OM [25] + TIM | $80.64 \pm 0.34$ | $\textbf{89.39} \pm \textbf{0.39}$ |
| ADV-CE + TIM | $\textbf{80.75} \pm \textbf{0.25}$ | $88.97 \pm 0.14$ |

## 4.1 Comparison with Baselines

We present the performance of ADV-CE, ADV-SVM with a WRN feature extractor and compare them with existing works in Table 1. The results of baselines are from [19], [26] and [13]. We observe in Table 1 that, in all datasets, ADV achieves the best performance in both 1-shot and 5-shot tasks. Notable, ADV outperforms the best competitor by at least 2% in 1-shot tasks. The extraordinary performance of ADV in 1-shot tasks indicates its effectiveness to exploit the information of the unlabeled data from the query set to provide a more accurate distribution estimate to the true distribution of each class, addressing the issue that the empirical distribution estimated from the support set may deviate significantly from the true distribution.

We also present the experimental results regarding comparison with more SOTA algorithms, including Transductive Information Maximation (TIM) [3], LaplacianShot [49] and Obliquer Manifold (OM) [25]. We compare with these 3 methods because they are all independent of the pre-training of the base model and do not require training any extra module in the meta-training phase. Note that the performance of few-shot learning is affected by both the base model obtained in the meta-training phase and the fine-tuning strategy in the meta-testing phase. For a fair comparison in the meta-testing phase, the base model should be the same for different methods. We feed all 4 methods the same fixed WRN feature extractor and the same 10,000 meta-testing tasks. As TIM and LaplacianShot use the same pre-trained weights of a feature extractor, we use these pre-trained weights for all of the 4 methods. No data augmentation is applied to images in meta-testing tasks. From the results in Table 2, we observe that ADV-CE outperforms all baselines in 1-shot learning and obtains the second best performance in 5-shot learning, which is slightly lower than TIM.

The major contributions of TIM lie in regularization on query sets, while ours can be deemed as regularization on support sets, which means its contributions are orthogonal to our contributions. We are more interested in integrating TIM to further boost the proposed method. When integrating TIM with ADV-CE and applying it to the WRN feature extractor pre-trained by [22] on miniImageNet, ADV-CE+TIM achieves competitive performance to the SOTA methods as presented in Table 3, where the performances of the SOTA methods are reported in the original paper.

## 4.2 Visualization of Vicinal Distributions

The vicinal risk minimization principle in the proposed method is built on the vicinal distributions obtained by a lazy random walk. Therefore, the performance of ADV depends on the exactness of the vicinal distribution estimation. Here, we visualize the vicinal distributions to evaluate their exactness. We run the proposed lazy random walk algorithm on two 5-way 1-shot tasks randomly constructed on the meta-testing set of miniImageNet, generate 500 samples from each vicinal distribution, and perform t-SNE algorithm on all data. The visualization results are presented in Figure 2.

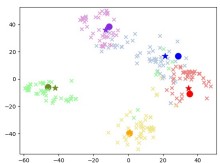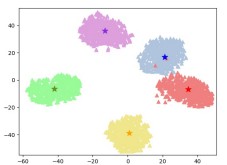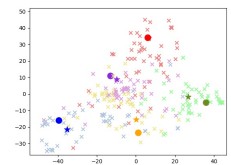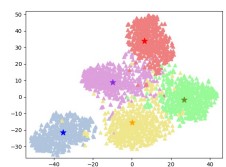

Figure 2: t-SNE visualization of the vicinal distribution and the true distribution. The first and second subfigures show the original data and the vicinal distributions for a task. The third and fourth subfigures show the results for another task. Different colors represent different classes. The '•' represents the support samples; '×' represents the query samples; '★' represents the mean of the vicinal distributions; and '▲' represents the samples generated from the vicinal distribution.

Table 4: Performance with different backbones on 5-way 1-shot and 5-way 5-shot classification accuracy (%) on miniImageNet with 95% confidence intervals.

| Backbone | CONV | | ResNet | | WRN | |
|---|---|---|---|---|---|---|
| Methods | 1-shot | 5-shot | 1-shot | 5-shot | 1-shot | 5-shot |
| Kernel SVM | $55.14 \pm 0.62$ | $68.37 \pm 0.51$ | $57.85 \pm 0.63$ | $73.47 \pm 0.51$ | $64.77 \pm 0.44$ | $80.79 \pm 0.32$ |
| ADV-SVM | $59.02 \pm 0.70$ | $\mathbf{72.13 \pm 0.53}$ | $64.27 \pm 0.74$ | $76.37 \pm 0.54$ | $72.18 \pm 0.48$ | $84.38 \pm 0.30$ |
| CE | $54.41 \pm 0.61$ | $70.74 \pm 0.52$ | $56.87 \pm 0.65$ | $76.73 \pm 0.49$ | $65.38 \pm 0.52$ | $83.33 \pm 0.30$ |
| ADV-CE | $\mathbf{59.22 \pm 0.68}$ | $72.07 \pm 0.51$ | $\mathbf{65.03 \pm 0.74}$ | $\mathbf{77.12 \pm 0.52}$ | $\mathbf{74.63 \pm 0.48}$ | $\mathbf{86.12 \pm 0.29}$ |

We can observe from Figure 2 that the vicinal distribution generated by ADV covers most of the query samples of the same class. This verifies that the lazy random walk module in ADV can provide a good estimate of the vicinal distribution, which overlaps with the ground truth distribution. By optimizing the expected loss over the vicinal distributions, the model trained by ADV has a smaller probability of overfitting the support samples and provides better generalization to the query set.

## 4.3 Effect of the Network Architecture of the Feature Extractor

The proposed algorithm ADV is compatible with any pre-trained feature extractor. Here, we validate the effectiveness of ADV on three feature extractors with different backbones, which are trained by different meta-learning methods. We consider WideResNet (WRN) [46], ResNet-12 [11] and a shallow network of 4 convolutional blocks (CONV) [2]. We meta-train the WRN, ResNet, CONV following [22], [19] and [2], respectively.

We report the performance of a kernel SVM classifier and a linear classifier trained by cross-entropy (denoted by CE) with and without ADV. The results in Table 4 show that the performance improvement of ADV is consistent for three feature extractors. This verifies that the effectiveness of ADV is independent of the feature extractor. It is worth noting that the larger performance gains are observed on the better feature extractors. For example, the overall performance of WRN outperforms CONV, and the performance boost of ADV-CE over CE with WRN in 1-shot tasks is 10.5%, which is larger than the boost with CONV (4.8%). This observation suggests that 1) even with a superior feature extractor that well preserves the similarity between data in feature space, the classifier trained from a limited number of high-quality support data still overfits the data; 2) for a better feature extractor, ADV can exploit more information from the similarity between data to estimate the vicinal distribution of support data and alleviate the overfitting.

## 4.4 Ablation Study

We perform an ablation study on ADV regarding the gain of the VRM principle. To do this, we train a vanilla linear classifier with cross-entropy loss (denoted by CE), a linear classifier with Mixup vicinal distribution [47] (denoted by CE-Mixup), a linear classifier with Delta vicinal distribution (denoted by ADV-CE-Mean) whose mean locates on the same mean estimate by our lazy random walk module and whose variance is zero, i.e., $P_{z_i}(\tilde{z}) = \delta_{\mu_i}(\tilde{z})$, and compare them with ADV-CE. We also train a multi-class SVM classifier, a multi-class kernel SVM classifier, and a kernel SVM classifier with

Table 5: Ablation Study of the proposed ADV-CE algorithm on 5-way 1-shot and 5-way 5-shot classification accuracy (%) on miniImageNet, CUB and CIFAR-FS with 95% confidence intervals.

| Methods | miniImageNet 5-way | | CUB 5-way | | CIFAR-FS 5-way | |
|---|---|---|---|---|---|---|
| | 1-shot | 5-shot | 1-shot | 5-shot | 1-shot | 5-shot |
| CE | 65.38 ± 0.52 | 83.33 ± 0.30 | 79.40 ± 0.46 | 90.36 ± 0.24 | 74.06 ± 0.47 | 87.74 ± 0.32 |
| CE-Mixup | 65.53 ± 0.43 | 82.77 ± 0.30 | 79.02 ± 0.45 | 90.14 ± 0.25 | 74.99 ± 0.47 | 87.45 ± 0.32 |
| ADV-CE-Mean | 72.37 ± 0.49 | 84.79 ± 0.31 | 83.04 ± 0.46 | 90.61 ± 0.25 | 80.56 ± 0.47 | 88.11 ± 0.31 |
| ADV-CE | **74.63 ± 0.48** | **86.12 ± 0.29** | **84.20 ± 0.43** | **91.54 ± 0.22** | **82.02 ± 0.48** | **88.25 ± 0.31** |
| SVM | 64.33 ± 0.44 | 83.57 ± 0.30 | 78.46 ± 0.45 | 90.93 ± 0.23 | 74.29 ± 0.46 | 87.31 ± 0.32 |
| Kernel SVM | 64.77 ± 0.44 | 80.79 ± 0.32 | 77.69 ± 0.45 | 88.96 ± 0.27 | 74.03 ± 0.46 | 86.28 ± 0.34 |
| ADV-SVM-Mean | 65.33 ± 0.43 | 81.36 ± 0.32 | 80.97 ± 0.48 | 90.94 ± 0.23 | 80.21 ± 0.48 | 87.37 ± 0.33 |
| ADV-SVM | **72.18 ± 0.48** | **84.38 ± 0.30** | **83.25 ± 0.44** | **91.23 ± 0.23** | **81.12 ± 0.46** | **87.41 ± 0.33** |

Delta vicinal distribution (denoted by ADV-SVM-Mean). The training of multi-class SVM classifiers all follows [7]. We report the performance on three benchmarks in Table 5.

The results in Table 5 show that the performance of CE-Mixup only competes with CE in miniImageNet and CIFAR-FS, and is even inferior to CE on the CUB dataset. This suggests that the Mixup vicinal distribution fails to improve the generalization given a limited number of data. In contrast, the vicinal distribution obtained by the lazy random walk algorithm is a more accurate estimate of the true distribution, as evidenced by the fact that the performance of ADV-CE-Mean and ADV-CE is better than CE and CE-Mixup. To be concrete, the performance boost of ADV-CE-Mean over CE demonstrates that the mean of the vicinal distribution, which lies not on the original sample but on a weighted average of the data passed through by the lazy random walk, is closer to the ground-truth distribution. The performance improvement of ADV-CE over ADV-CE-Mean demonstrates that the variance of the vicinal distribution helps to alleviate the overfitting problem.

As shown in Table 5, the performance of Kernel SVM is worse than SVM in most cases. The reason may be that the features extracted by WRN are linearly separable and the non-linear kernel technique does not help to further improve the separability. Although the performance of Kernel SVM is inferior to SVM, we observe a performance boost of ADV-SVM, which is built on Kernel SVM. This indicates that learning from the vicinal distributions that are close to the true distribution would sufficiently improve the performance. We also observe the performance gain of ADV-SVM over ADV-SVM-DD, again evidencing the importance of the variance of vicinal distributions in alleviating the overfitting.

## 5 Conclusion and Limitations

In this paper, we discuss the overfitting issue in few-shot learning, which is caused by the deviation of the empirical distribution from the ground-truth distribution. To alleviate the overfitting, we revisit the Vicinal Risk Minimization principle in deep few-shot learning and propose to harness unlabeled data and a lazy random walk algorithm to generate adaptive statistical parameters for the vicinal distribution of each training sample. Following the VRM principle, we learn the model by optimizing the resulting expected vicinal loss. We derive the vicinal loss for linear classifiers trained with cross-entropy loss and for SVM classifier and develop the resulting Adaptive Vicinal classifiers as ADV-CE and ADV-SVM, respectively. We evaluate the performance of ADV-CE and ADV-SVM on three benchmarks and validate the effectiveness of ADV in mitigating overfitting and improving the generalization of classifiers learned from a limited number of samples. Currently, our proposed ADC is compatible with CE and SVM only. We will extend it to other loss functions. For the vicinal Cross-Entropy loss, we optimize an approximation obtained by second-order Taylor expansion. However, the approximation error can be large. Thus, the performance of ADV-CE may be suboptimal and we can further improve it by deriving a better approximation for the vicinal Cross-Entropy loss.

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
