# A  Derivation of vicinal kernel functions

Here, we provide the derivation of vicinal kernel functions omitted in Section 3.2.2.

The one-side vicinal kernel $\mathcal{Q}(z, z_i)$ is derived as

$$
\begin{aligned}
\mathcal{Q}(z_i, z) &:= \mathbb{E}_{\tilde{z}_i \sim \mathcal{N}(\mu_i, \Sigma_i)} \mathcal{K}(\tilde{z}_i, z) = \int \mathcal{K}(z_i, \tilde{z}_i) \, \mathrm{d}P_{z_i}(\tilde{z}_i) \\
&= (2\pi)^{-\frac{d}{2}} |\Sigma_i|^{-\frac{1}{2}} \int \exp\left(-\frac{\|z - z'\|^2}{2\sigma^2}\right) \exp\left(-\frac{1}{2}(z' - \mu_i)^\top \Sigma_i^{-1}(z' - \mu_i)\right) \, \mathrm{d}z' \\
&= \frac{|A(\sigma^2)|^{\frac{1}{2}}}{|A(\sigma^2) + \Sigma_i|^{\frac{1}{2}}} \exp\left(-\frac{1}{2}(z - \mu_i)^T \left[A(\sigma^2) + \Sigma_i\right]^{-1}(z - \mu_i)\right).
\end{aligned}
$$

And the two-side vicinal kernel $\mathcal{M}(z_i, z_j)$ is derived as

$$
\begin{aligned}
\mathcal{M}(z_i, z_j) &:= \mathbb{E}_{\tilde{z}_i \sim \mathcal{N}(\mu_i, \Sigma_i)} \mathbb{E}_{\tilde{z}_j \sim \mathcal{N}(\mu_j, \Sigma_j)} \mathcal{K}(\tilde{z}_i, \tilde{z}_j) = \int \int \mathcal{K}(\tilde{z}_i, \tilde{z}_j) \, \mathrm{d}P_{z_i}(\tilde{z}_i) \, \mathrm{d}P_{z_j}(\tilde{z}_j) \\
&= (2\pi)^{-\frac{d}{2}} |\Sigma_i|^{-\frac{1}{2}} |\Sigma_j|^{-\frac{1}{2}} \\
&\quad \int \int \left\{\exp\left(-\frac{\|z - z'\|^2}{2\sigma^2}\right) \exp\left(-\frac{1}{2}(x' - \mu_i)^\top \Sigma_i^{-1}(x' - \mu_i)\right) \exp\left(-\frac{1}{2}(x - \mu_j)^\top \Sigma_j^{-1}(x - \mu_j)\right)\right\} \mathrm{d}z \mathrm{d}z' \\
&= \frac{|A(\sigma^2)|^{\frac{1}{2}}}{|A(\sigma^2) + \Sigma_i + \Sigma_j|^{\frac{1}{2}}} \exp\left(-\frac{1}{2}(\mu_i - \mu_j)^T \left[A(\sigma^2) + \Sigma_i + \Sigma_j\right]^{-1}(\mu_i - \mu_j)\right)
\end{aligned}
$$

The derivation follows the convolution integrals of normal distribution functions and is simplified by the assumption that $A(\sigma^2), \Sigma_i, \Sigma_j$ are diagonal.

# B  Introduction to Datasets in Experiments

We evaluate the proposed method on *mini*ImageNet [39], CUB [40] and CIFAR-FS [2].

***mini*ImageNet** is a standard benchmark for few-shot image classification. It consists of 100 classes from ImageNet dataset [30]. Each class contains 600 images of size 84×84. These classes are split into 64, 16, and 20 classes for meta-training, meta-validation, and meta-testing respectively [28].

**CUB** contains 200 classes with a total of 11,788 images of size $84 \times 84$. Following previous works [5], the base, validation, and novel split are 100, 50, and 50 classes respectively.

**CIFAR-FS** is a variant of the CIFAR-100 dataset used for few-shot classification. It contains 100 classes, each with 600 images of $32 \times 32$ pixels. The classes are randomly split into 64, 16, and 20 for meta-training, meta-validation, and meta-testing respectively.