# OpenReview forum: "Improving Task-Specific Generalization in Few-Shot Learning via Adaptive Vicinal Risk Minimization"
_NeurIPS.cc/2022/Conference — NeurIPS 2022 Accept_

### Official Review · Reviewer_oXzN · 2022-06-21

**Rating:** 6
**Confidence:** 4
**Soundness:** 3 good
**Presentation:** 2 fair
**Contribution:** 3 good

**Summary:**

The paper proposes to use vicinal loss function in few shot classification to increase the generalization capability.  The vicinal loss is associated with  some prior distributions over the features, which is adaptively obtained via random walk algorithm with the assistance of a set of unlabeled examples.

**Questions:**

* The reviewer believe the statement in L133-L135 is technically not correct, can the authors address this concern?
* The reviewer believe the vicinal loss is the same as using prior distribution on the features, can the authors clarify?
* The details of the ablation study is not very clearly presented in the paper.  For example, it is not clear to the reviewer what the delta vicinal distribution is and how it is different from the distribution used in ADV-CE.
* Unlabeled data.
    * The paper does not provide any information on the size of unlabeled data used in the experiment.
    * Does the proposed method give higher classification accuracy with more unlabeled data? This should be shown in the ablation study.

**Limitations:**

Some limitations of the work has been briefly mentioned by the authors. The limitations and questions of the paper the reviewer had is given in Questions section.

**Strengths And Weaknesses:**

* Originality: The task of few shot learning and the techniques of vicinal loss and random walk algorithms are not new. To the best of the reviewer’s knowledge, the use of these techniques in few shot learning is novel. However, there is a lack of comparison with other methods（eg [15]）use additional unlabeled data.
* Quality: All claims are supported by the experimental results and the work is complete. The reviewer do not agree with the statement that the proposed method can beat SOTA.
* Clarity: The idea of the paper is presented clearly. However, there are many typos, especially in the equations.
	* typos (an incomplete list):
		* L110: citation is not right
		* L126: should be “the” not “tge”
		* L130: model is $F(x;\theta)$ not $f_{\theta}(x)$
		* L140: model parameter is $\theta$ not $w$
		* L146: the density function is a function of $\tilde z$ on the LHS but does not depend on $\tilde z$ on the RHS
		* L71: the expectation is with respect to z not $\bar z_i$
	* technical error:
		* L133-L135: Although the form of $dP_{\delta}(x,y)$ is given in the original paper, this is technically not right. The decomposition is saying that x and y are independent. The vicinal loss from a Bayesian point of view can be seen as adding a prior on the predictor x
* Significance: The paper proposes to use a loss function that is different from the conventional empirical loss under few shot learning setting. The motivation for this approach may be insightful for developing  new methods in few shot learning. The results, while not earthshaking, may be of practical use.

---

> ### Author Response · Authors · 2022-08-02
> **Response to Reviewer oXzN (Part I)**
>
> We sincerely appreciate your comments on this paper. You may find our response below for your concerns. We would really appreciate it if you could let us know if you have any further concerns.
>
> #### Q1: Although the form of $P_{\delta}(x, y)$ is given in the original paper, this is technically not right. The decomposition is saying that x and y are independent.
>
> > There might be some misunderstanding here. The form of $d P_{\delta} (x, y)$ is technically correct. The decomposition of $d P_{\delta}​(x,y)$ into Dirac delta functions does not mean that $x$ and $y$ are independent.
> >
> > - Firstly, the empirical density is estimated by the training samples $\{x_i, y_i\}$. So we have  $d P_{\delta} (x, y) =\frac{1}{N} \sum_{i=1}^N d P_{\delta}(x_i, y_i)$ .
> >
> > - By the property of Dirac delta function, we have $d P_{\delta}(x_i, y_i) = \delta_{[x_i, y_i]}([x, y]) = \delta_{x_i}(x) \delta_{y_i}(y)$, which means that for any $x \neq x_i$ or $y \neq y_i$, the density is zero. However, at the point $(x_i,y_i)$, $x_i$ and $y_i$ are still highly dependent in $d P_{\delta}(x_i, y_i)$.
> >
> > - Improvement of the density estimate is obtained by replacing $\delta_{x_i}(x)$ by a vicinal distribution $P_{x_i}(x)$ around $x_i$. It means that for any $x$ in the vicinity of $x_i$, i.e., whose means $x$ is close to $x_i$, its label is $y_i$. Obviously, $x\sim P_{x_i}(x)$ and $y_i$ are highly correlated.
>
> #### Q2: The reviewer believes the vicinal loss is the same as using a prior distribution on the features, can the authors clarify?
>
> > - Yes. The vicinal distribution of the data can be viewed as a prior distribution of the features. Concretely, we use **an instance-wise non-isotropic Gaussian vicinal distribution / prior** obtained by performing a lazy random walk among the unlabeled data for each instance in the support set.
>
> #### Q3: it is not clear to the reviewer what the delta vicinal distribution is and how it is different from the distribution used in ADV-CE.
>
> > - The vicinal distribution used in ADV-CE is a Gaussian distribution $P_{x_i}(x)=\mathcal{N}(\mu_i, \Sigma_i)$, where we obtain both $\mu_i$ and $\Sigma_i$ by the lazy random walk algorithm as introduced in Sec. 3.3. The Delta vicinal distribution defined as $P_{x_i}(x)=\delta_{\mu_i}(x)$, in contrast, only estimates the $\mu_i$ by the lazy random walk algorithm.
> >
> > - The key difference between these two distributions, therefore, is that **the variance of the Delta vicinal distribution is zero**. In the ablation study in Table 3, the performance gain of using Gaussian vicinal distribution over using Delta vicinal distribution achieves about 2% in the miniImageNet dataset. This underlines the importance of estimating the true variance in the vicinal distribution.
>
> #### Q4: The size of unlabeled data used in the experiment.
>
> > - The size of unlabeled data is the same as the number of query data. It is 15 per class following the standard few-shot learning setting.
>
> #### Q5: Does the proposed method give higher classification accuracy with more unlabeled data?
>
> > - We have followed the reviewer's suggestion to conduct an experiment on miniImageNet by **varying different numbers of query samples** (equivalent to **unlabeled samples** in our case). We report the results in the table below.
> >
> >   - The performance of ADV-CE indeed increases with more unlabeled samples, though it tends to converge if the number of query samples per class gets sufficiently large (e.g., > 30).
> >
> >   - As for CE, its performance barely changes with the number of query data, explained by the fact that it is not transductive.
> >
> > | number of query data | 5            | 15           | 30           | 50           |
> > |----------------------|--------------|--------------|--------------|--------------|
> > | 1 shot: CE           | 65.58 ± 0.50 | 65.38 ± 0.52 | 65.33 ± 0.40 | 65.37 ± 0.39 |
> > | 1 shot: ADV-CE       | 72.94 ± 0.59 | 74.63 ± 0.48 | 76.39 ± 0.43 | 76.72 ± 0.42 |
> > | 5 shot: CE           | 83.22 ± 0.37 | 83.33 ± 0.30 | 83.24 ± 0.28 | 83.23 ± 0.26 |
> > | 5 shot: ADV-CE       | 84.93 ± 0.38 | 86.12 ± 0.29 | 86.93 ± 0.24 | 87.15 ± 0.23 |

---

> > ### Comment · Reviewer_oXzN · 2022-08-05
> > **Thank you for your response**
> >
> > I am happy with most of your response though I am still not convinced by your answer to Q1. Anyway, this is not a big deal in terms of the general idea of the paper.
> >
> > Although the techniques are not new, the application of vinical loss in few shot learning is interesting. I would be interested to see more discussions on the choice of the prior since the approach that you have picked *"an instance-wise non-isotropic Gaussian vicinal distribution / prior"*.
> >
> > I agree with other reviewers that the paper definitely needs to be well proofread before it gets ready for publication. I will keep my rating same as before.

---

> > > ### Author Response · Authors · 2022-08-07
> > > **Response to Reviewer oXzN**
> > >
> > > Thanks again for your valuable comments. We have double-checked the paper, corrected typos, added more explanation to make it clearer, and submitted the revision.
> > >
> > > Q7: more discussions on the choice of the prior.
> > >
> > > > - In our method, the feature of each instance admits a vicinal distribution. So the prior is instance-wise.
> > > > - We choose Gaussian vicinal distribution for both ADV-CE and ADV-SVM for the following reasons.
> > > >   - In ADV-CE, the approximated expected vicinal loss function involves only the mean and variance of the vicinal distribution. So we consider Gaussian vicinal distribution, which is parameterized by mean and variance and has zero higher order moments.
> > > >   - In ADV-SVM, Gaussian vicinal distribution is closely related to RBF kernel. The derivation of the closed form solution for vicinal SVM depends on using a Gaussian vicinal distribution.
> > > >   - The Gaussian prior provides a vicinal distribution of sufficient information in the few-shot learning.
> > > > - The isotropic Gaussian vicinal distribution, whose covariance matrix is represented by the simplified matrix $\Sigma = \sigma^2 I$, assumes different dimensions are independent and have the same variance. But in practice, different dimensions of the feature may be correlated and has different variances. Therefore, we choose non-isotropic Gaussian vicinal distribution.

---

> ### Author Response · Authors · 2022-08-02
> **Response to Reviewer oXzN (Part II)**
>
> #### Q6: Comparison with other methods that use additional labeled data.
> > - The performance of few-shot learning is affected by both the base model obtained in the pre-training phase (meta-training phase) and the fine-tuning strategy in the fine-tuning phase (meta-testing phase). **For a fair comparison in the meta-testing phase, the base model should be the same for different methods.**
> >
> > - Therefore, we compared ADV-CE with **3 SOTA methods**: Transductive Information Maximation (TIM) [R1], LaplacianShot [R2], and Obliquer Manifold (OM) [R5].
> >
> >   - The reason why we select these 3 methods is that they are all independent of the pre-training of the base model and do not require training any extra module in the pre-training /meta-training phase.
> >
> >   - **For a fair comparison**, we feed all methods the same fixed WRN feature extractor and the same 10,000 meta-testing tasks. As TIM and LaplacianShot use the same pre-trained weights of a feature extractor, we use these pre-trained weights for all of the 4 methods. No data augmentation is applied to images in meta-testing tasks.
> >
> > - We present the results in the table below. We observe that ADV-CE outperforms all baselines in 1-shot learning and obtains the second best performance in 5-shot learning, which is slightly lower than TIM.
> >
> > | Method            | 1-shot       | 5-shot       |
> > |-------------------|--------------|--------------|
> > | TIM [R1]           | 74.36 ± 0.26 | 84.58 ± 0.15 |
> > | LaplacianShot [R2] | 69.10 ± 0.23 | 80.86 ± 0.15 |
> > | OM [R5]            | 68.67 ± 0.23 | 83.42 ± 0.15 |
> > | ADV-CE            | 74.92 ± 0.25 | 84.53 ± 0.16 |
> >
> > - Note that the major contributions of TIM lie in regularization on query sets, though ours can be deemed as regularization on support sets, **which means its contributions are orthogonal to our contributions**. We are more than interested in integrating TIM to further boost ours. When integrating TIM with ADV-CE and applying it on WRN feature extractor pre-trained by [19] on miniImageNet, ADV-CE+TIM achieves competitive performance to SOTA methods in the following table, where the performances of SOTA methods are reported in the original paper.
> >
> >   | Method             | 1-shot       | 5-shot       |
> >   | ------------------ | ------------ | ------------ |
> >   | TIM [R1]           | 77.8         | 87.4         |
> >   | LaplacianShot [R2] | 74.86 ± 0.19 | 84.13 ± 0.14 |
> >   | [R3]               | 80.04        | 87.64        |
> >   | [R4]               | 76.84        | 84.36        |
> >   | OM [R5]             | 80.64 ± 0.34 | 89.39 ± 0.39 |
> >   | ADV-CE+TIM         | 80.75 ± 0.25 | 88.97 ± 0.14 |
> >
> > [R1] Boudiaf, Malik, et al. "Information maximization for few-shot learning." NeurIPS 2020.
> >
> > [R2] Ziko, Imtiaz, et al. "Laplacian regularized few-shot learning." ICML 2020.
> >
> > [R3] Cui, Wentao, and Yuhong Guo. "Parameterless transductive feature re-representation for few-shot learning." ICML 2021.
> >
> > [R4] Lee, Dong Hoon, and Sae-Young Chung. "Unsupervised Embedding Adaptation via Early-Stage Feature Reconstruction for Few-Shot Classification." ICML 2021.
> >
> > [R5] Qi, Guodong, et al. "Transductive Few-Shot Classification on the Oblique Manifold." ICCV. 2021.

---

### Official Review · Reviewer_WJab · 2022-06-25

**Rating:** 5
**Confidence:** 4
**Soundness:** 3 good
**Presentation:** 2 fair
**Contribution:** 2 fair

**Summary:**

In this paper, the distribution of support data is estimated using transition probability between all samples within a meta-test task (including labeled support data and unlabeled query data). A logistic regression or an SVM model is learned via vicinity risk minimizing based on the estimated distribution of support data. The expected vicinity loss is derived in the paper. The experiments indicate that the proposed method outperforms some few-shot learning methods.

**Questions:**

Typos in line 110.

What is "tge" in line 126?

According to line 210 and 211, it is not clear how to compute $u_i$ and $\Sigma_i$. The definition of $V_i$ is not given in line 211. Is it a typo? Is it $v_i$ or $V_i$? It is better to give equations about how to compute $u_i$ and $\Sigma_i$.

What is the value of $N_u$?

**Limitations:**

The author discussed limitations in the paper.

**Strengths And Weaknesses:**

The expected vicinity loss is derived.

Estimate the distribution of support data using weighed unlabeled data, where the weight is based on the transition probability in lazy random walk.


Weaknesses:

Some typos in the manuscript. Some technical details are missing. See the next section for details.

The proposed method is a transductive few-shot learning method as it uses unlabeled data in meta-test tasks. Therefore, it is important to compare to transductive few-shot learning methods with the same backbone. However, the comparison to transductive methods in Table 1 is not fair. For example, the reported results of some methods (e.g. FEAT) are based on shallow backbones (4-layer CNN). It is better to reproduce those methods with the same backbone (WRN), especially since the code is available online for those methods.

In addition, some recent transductive few-shot methods [1, 2, 3, 4, 5] achieve stronger performance using the same backbone (WRN). The proposed method should be compared with those methods to show the effectiveness of adaptive vicinity risk minimization. Since the authors claim that the proposed method outperforms SOTA few-shot learning methods, it is necessary to compare the true SOTA methods in a fair manner.

[1] Boudiaf, Malik, et al. "Information maximization for few-shot learning." Advances in Neural Information Processing Systems 33 (2020): 2445-2457.

[2] Ziko, Imtiaz, et al. "Laplacian regularized few-shot learning." International Conference on Machine Learning. PMLR, 2020.

[3] Cui, Wentao, and Yuhong Guo. "Parameterless transductive feature re-representation for few-shot learning." International Conference on Machine Learning. PMLR, 2021.

[4] Lee, Dong Hoon, and Sae-Young Chung. "Unsupervised Embedding Adaptation via Early-Stage Feature Reconstruction for Few-Shot Classification." International Conference on Machine Learning. PMLR, 2021.

[5] Qi, Guodong, et al. "Transductive Few-Shot Classification on the Oblique Manifold." Proceedings of the IEEE/CVF International Conference on Computer Vision. 2021.

---

> ### Author Response · Authors · 2022-08-02
> **Response to Reviewer WJab (Part I)**
>
> We thank the reviewer for the valuable feedback. We address your concerns below point by point. Please kindly let us know whether you have any further concerns.
>
> #### Q1: Comparison with transductive few-shot learning methods under the same backbone. Reproduce FEAT with WRN.
>
> > - 1\) In Table 2, we have already provided the performance comparison of the proposed method using the backbone of CONV and ResNet-12. Therefore, by **joining Table 2 and Table 1**, we observe that **ours using the 4-layer CNN (i.e., CONV) still achieves the highest 1-shot classification accuracy (59.22)** and competitive 5-shot classification accuracy (72.07) among all the tranductive methods that use CONV (including FEAT).
> >
> > - 2\) We have also run the codes of FEAT with the WRN backbone and evaluated on miniImageNet. The results reported in the table below again demonstrate the **superiority of ours** over the competitive tansductive baseline **FEAT equipped with WRN**.
> >
> > | Method |1 shot | 5 shot |
> > | --| --| --|
> > | FEAT (WRN) | 71.80 ± 0.22  | 79.81 ± 0.16     |
> > | ADV-CE (WRN) | **74.63 ± 0.48** | **86.12 ± 0.29** |
>
> #### Q2: Comparison with the true SOTA methods in a fair manner.
>
> > - We thank the reviewer for providing these 5 SOTA methods [R1-R5]. Note that the performance of few-shot learning is affected by both the base model obtained in the pre-training phase (meta-training phase) and the fine-tuning strategy in the fine-tuning phase (meta-testing phase). **For a fair comparison in the meta-testing phase, the base model should be the same for different methods.**
> >
> > - Therefore, we have followed the reviewer's suggestion by comparing ADV-CE with **3 SOTA methods**: Transductive Information Maximation (TIM) [R1], LaplacianShot [R2], and Obliquer Manifold (OM) [R5].
> >
> >   - The reason why we select these 3 methods is that they are all independent of the pre-training of the base model and do not require training any extra module in the pre-training /meta-training phase.
> >
> >   - Instead, we exclude [R3] because (1) it requires training an extra feature re-representation layer, (2) it requires to pre-train the model with self-supervised learning, and (3) it does not open-source the codes. We similarly exclude [R4] because it requires training an extra module to extract task-useful features.
> >
> >   - **For a fair comparison**, we feed all methods the same fixed WRN feature extractor and the same 10,000 meta-testing tasks. As TIM and LaplacianShot use the same pre-trained weights of a feature extractor, we use these pre-trained weights for all of the 4 methods.  No data augmentation is applied to images in meta-testing tasks.
> >
> > - We present the results in the table below. In 1-shot learning, ADV-CE achieves the best performance and in 5-shot learning, ADV-CE also achieves competitive performance, which is just 0.05% lower than the best performance obtained by TIM. This verifies that the vicinal distribution is an accurate approximation of the true distribution and helps to improve the generalization..
> >
> > | Method  | 1-shot | 5-shot |
> > |--|--|--|
> > | TIM [R1] | 74.36 ± 0.26 | 84.58 ± 0.15 |
> > | LaplacianShot [R2] | 69.10 ± 0.23 | 80.86 ± 0.15 |
> > | OM [R5] | 68.67 ± 0.23 | 83.42 ± 0.15 |
> > | ADV-CE | 74.92 ± 0.25 | 84.53 ± 0.16 |
> >
>
> > - Note that the major contributions of TIM lie in regularization on query sets, though ours can be deemed as regularization on support sets, **which means its contributions are orthogonal to our contributions**. We are more than interested in integrating TIM to further boost ours. When integrating TIM with ADV-CE and applying it on WRN feature extractor pre-trained by [19] on miniImageNet, ADV-CE+TIM achieves competitive performance to SOTA methods in the following table, where the performances of  SOTA methods are reported in the original paper.
> >
> >   | Method | 1-shot | 5-shot |
> >   | -- | -- | -- |
> >   | TIM [R1]  | 77.8 | 87.4 |
> >   | LaplacianShot [R2] | 74.86 | 84.13 |
> >   | [R3] | 80.04 | 87.64 |
> >   | [R4] | 76.84 | 84.36 |
> >   | OM[R5] | 80.64 | 89.39 |
> >   | ADV-CE+TIM | 80.75 | 88.97 |
> >
> > [R1] Boudiaf, Malik, et al. "Information maximization for few-shot learning." NeurIPS 2020.
> >
> > [R2] Ziko, Imtiaz, et al. "Laplacian regularized few-shot learning." ICML 2020.
> >
> > [R3] Cui, Wentao, and Yuhong Guo. "Parameterless transductive feature re-representation for few-shot learning." ICML 2021.
> >
> > [R4] Lee, Dong Hoon, and Sae-Young Chung. "Unsupervised Embedding Adaptation via Early-Stage Feature Reconstruction for Few-Shot Classification." ICML 2021.
> >
> > [R5] Qi, Guodong, et al. "Transductive Few-Shot Classification on the Oblique Manifold." ICCV. 2021.

---

> ### Author Response · Authors · 2022-08-02
> **Response to Reviewer WJab (Part II)**
>
> #### Q3: It is not clear how to compute $\mu_i$ and $\Sigma_i$
>
> > - 1\) $V_i$ is a typo. It should be $v_i$.
> >
> > - 2\) The computation: $\mu_i = \frac{1}{2} z_i + \frac{1}{2} \sum_{j=1}^{N_u} v_j u_j$ and $\Sigma_i = \frac{1}{2}(z_i-\mu_i)(z_i-\mu_i)^\top + \frac{1}{2}\sum_{j=1}^{N_u} v_j (u_j - \mu_i)(u_j-\mu_i)^\top$
>
> #### Q4: What is the value of $N_u$?
>
> > - $N_u$ is the number of unlabeled samples, which is the same as the number of query samples. It is 75 (15 query samples per class multiplying 5 classes)  in the experiments.
>
> #### Q5: Typos
>
> > - 1\) Typos in Line 110. A: It is a failed author citation. It should be Chapelle et al. [3].
> >
> > - 2\) What is "tge" in line 126?  A: It should be "the".
> >
> > - We fix the typos in the revised manuscript.

---

> ### Author Response · Authors · 2022-08-05
> **Please let us know if you have any further questions.**
>
> Hi Reviewer WJab,
>
> We would like to follow up to see if our response addresses your concerns or if you have any further questions. We would really appreciate the opportunity to discuss this further if our response has not already addressed your concerns. Thank you again!

---

> ### Comment · Reviewer_WJab · 2022-08-05
> **After rebuttal**
>
> Thank you for the response from the authors. It addresses my concerns. I will raise my score.

---

### Official Review · Reviewer_kfYN · 2022-07-11

**Rating:** 4
**Confidence:** 5
**Soundness:** 2 fair
**Presentation:** 2 fair
**Contribution:** 2 fair

**Summary:**

This submission points out that, in few-shot learning, a limited number of support data can hardly represent the true distribution, which results in poor generalization performance of the empirical risk minimization based methods. To tackle this problem, the authors propose to minimize the vicinal risk and estimate parameters of vicinal distribution of each sample by a random walk algorithm with additional unlabeled data. They demonstrate that their algorithm outperforms baseline methods on three datasets.

**Questions:**

Methods:
1. The pre-training phase is the meta-training phase? if not, the authors should detail the learning process of ADV in the meta-training and meta-testing phases.
2. The main problem is the difference between ADV-CE and ADV-SVM methods. They seem to be two separate branches, and ADV-CE significantly outperforms ADV-SVM. Therefore why provides AVD-SVM?
3. If i=j, the D_{ij}^{UU} = 0 in equation (11)?

Experiments:
1. How many query samples are used as unlabeled data to estimate the distribution parameters in experiments?  In general, the commonly used sample size is 15, and the more samples, the better the performance.
2. In lines 238-239, the authors propose that the support set can deviate from the true distribution. If there is a visualization to support this claim?
3. In the meta-testing phase, the meta-learning methods (e.g., MAML) are optimized by using 5-10 steps in the fine-tuning phase and the more steps have good performance [1]. However, ADV has 100 steps in the fine-tuning phases. if ADV also uses the same steps (5-10), how does it perform?
4. Only four transductive algorithms (TPN, FEAT, TEAM, SIB) are tested with the proposed method, it would be more convincing if more recent state-of-the-art algorithms can be considered.
5. The performance gain of the proposed method seems to diminish quickly from 1-shot to 5-shot settings. Does this mean in the case of 5 (or more)-shot setting, considering unlabeled test data with the proposed method could even adversely affect the performance?

Typos:
1. imporving --- imporve in the title
2. author? in the 110 line
3. 4.0.1 in the 222 line
4. ishas in the 254 line

[1] How to Train Your MAML to Excel in Few-Shot Classification

**Limitations:**

Yes, they discuss the limitations in Section 5.

**Strengths And Weaknesses:**

Strengths:
1. This paper points out an interesting problem for few-shot image classification.
2. The experimental evaluation is adequate, and the results convincingly support the main claims.

Weaknesses:
1. The paper is not very clear, and many aspects make the reviewer's understanding of the entire paper very confusing. Moreover, many typos should be carefully checked before submission.
2. The main concern is that the core idea is not innovative, similar to the incremental superposition of multiple algorithms.
3. Some experimental settings are unfair, making it impossible to measure the effectiveness of the method in terms of performance gains.
4. See questions for more details.

---

> ### Author Response · Authors · 2022-08-02
> **Response to Reviewer kfYN (Part I)**
>
> We sincerely appreciate your valuable comments on this paper. You may find our response below for your concerns. We would really appreciate it if you could let us know if you have any further concerns.
>
> #### Q1: Pre-training phase
>
> > - 1\) The proposed method focuses on **the meta-testing phase** and can be applied on top of any fixed feature extractor pre-trained / meta-trained on the base classes. This strategy follows existing works such as FreeLunch [39], TIM [R1], Laplacianshot [R2].  We just assume the existence of a pre-trained feature extractor / base model, regardless of how the feature extractor is pre-trained. In meta-testing phase, the feature extractor is fixed. We use the feature extractor to extract feature for all support data $\{z_i\}$ and query data $\{u_j\}$ and run Algorithm 1 for each meta-test task.
> >
> > - 2\) In the experiments, we have applied the proposed method on **3 different base models pre-trained by 3 different few-shot learning algorithms**. Concretely, we use WideResNet (WRN) pre-trained by [19], ResNet-12 pre-trained by [16] and 4-layer convolution network pre-trained [2] (as introduced in line 261.) The compatibility of our method with all of the 3 pre-trained feature extractors has been validated in Tab. 2. We also note that the results in Tab.1 and Tab.3 are obtained with the WRN pre-trained model.
> >
> > - We will add more pre-training details in the experiment.
> >
> > [R1] Boudiaf, Malik, et al. "Information maximization for few-shot learning." NeurIPS 2020.
> >
> > [R2] Ziko, Imtiaz, et al. "Laplacian regularized few-shot learning." ICML 2020.
>
> #### Q2: The difference between ADV-CE and ADV-SVM methods. Why provides AVD-SVM?
>
> > - 1\) The difference between ADV-CE and ADV-SVM lies in only the objective function that trains the classifier within each task -- ADV-CE taking the cross-entropy loss while ADV-SVM taking the max-margin loss. Estimation of the adaptive vicinal distribution in ADV-CE and ADV-SVM follows the same way.
> >
> > - 2\) We introduce ADV-CE and ADV-SVM to show the compatibility of the proposed ADV method with different classifiers. SVM has been one of the commonly adopted classifiers in few-shot algorithms such as MetaOptNet [16] and FreeLunch [39].
> >
> > - 3\) The slightly worse performance of ADV-SVM than ADV-CE in Tab.1/Tab.2/Tab.3 is attributed to the **inconsistency in the classifier used during pre-training and that used in meta-testing**.
> >
> >   - As stated in the response to Q1, we adopt the WRN pre-trained model which takes the CE loss to train the classifier during pre-training.
> >
> >   - If we use the ResNet-12 model pre-trained by MetaOptNet-SVM, where the classifier in pre-training is SVM, we observe from the following table that ADV-SVM outperforms ADV-CE on miniImageNet.
> >
> > | Methods | 1-Shot       | 5-shot       |
> > |---------|--------------|--------------|
> > | ADV-CE  | 64.71 ± 0.74 | 76.77 ± 0.52 |
> > | ADV-SVM | 64.95 ± 0.74 | 77.44 ± 0.48 |
> >
>
> #### Q3: $D_{ij}^{UU} = 0?$ if $i=j$ in Eqn. (11)
>
> > - In Eqn. (11), it is correct to set $D_{ij}^{UU} = \infty$ if $i=j$.
> >
> > - In the lazy random walk algorithm, we fix the lazy stay probability as $\beta$. By Eqn. (13), the stay probability is $\beta + (1-\beta) P_{jj}^{UU}$. Combining both, we expect $P_{jj}^{UU}=0$,  so as to deliberately set $D_{ij}^{UU}=\infty$ for $i=j$.
>
> #### Q4: How many query samples are used as unlabeled data?
>
> > - We do not use extra unlabeled data, and strictly follow the setting of few-shot learning where 15 query samples per class are regarded as the unlabeled data.
>
> #### Q5: Visualization to support the claim that the support set can deviate from the true distribution
>
> > - This claim has been supported by the **visualization in Fig 2 (Sec. 4.2)** where the 1-shot support data (denoted as dots) may deviate from the class mean of the true distribution (denoted as crosses). For example, in the first figure, the support data of the blue and red classes significantly deviate from the true class mean. In the third figure, almost all support data clearly deviate from their true class means.

---

> ### Author Response · Authors · 2022-08-02
> **Response to Reviewer kfYN (Part II)**
>
> #### Q6: If ADV also uses the same steps (5-10), how does it perform?
>
> > - 1\) Based the number of fine-tuning steps in the meta-testing phase, meta-learning algorithms can be categorized into (i) zero-step algorithms, e.g. metric based algorithm such as ProtoNet and MatchingNet; (ii) few-step algorithms, e.g., MAML and MetaSGD; and (iii) many-step algorithms, e.g., MetaOptNet[16], R2D2[2], and Free Lunch[39]. The proposed method falls into **the third category of many-step algorithms**, aiming to solve the objective to get a **near-optimal classifier by 100 steps**.  We would like to highlight that such a **comparison is fair**, as the baselines in Table 2 include several algorithms in the third category.
> >
> > - 2\) We show the results of 5-step ADV-CE on miniImageNet in the table below. By initializing the weights of the classifier with the prototypes of the support set, our ADV-CE achieves very **competitive performance by only 5 steps**. With more steps, the performance will slightly increase.
> >
> > | Methods         | 1-shot       | 5-shot       |
> > |-----------------|--------------|--------------|
> > | 100-step ADV-CE | 74.63 ± 0.48 | 86.12 ± 0.29 |
> > | 5-step ADV-CE   | 73.44 ± 0.49 | 85.42 ± 0.28 |
> > | 100 step CE     | 65.38 ± 0.52 | 82.03 ± 0.32 |
> > | 5 step CE       | 63.59 ± 0.48 | 79.21 ± 0.32 |
> >
>
> #### Q7: Comparison with more SOTA algorithms.
>
> > - Please refer to the answer to **Q2** of Reviewer **WJab**, where we provide comparison results with 3 new SOTA algorithms. The comparison is fair as the per-trained models for all algorithms are exactly the same and fixed.
>
> #### Q8: Explanation for why the performance gain of the proposed method seems to diminish quickly from 1-shot to 5-shot settings.
>
> > - 1\) Our method is not intended for many-shot problems. In the many-shot setting, the empirical distribution estimated by the support set closely approximates the true distribution. This means that estimating the vicinal distribution is less appealing, going against our motivation.
> >
> > - 2\) Even in the 5-shot setting, ADV-CE **outperforms CE and S2M2*** (one of the most competitive baselines in the 5-shot setting ) **by 2.79% and 3% on miniImageNet** with the WRN pre-trained model, which does not diminish.
> >
> > - 3\) We have conducted experiments of 10-shot and 15-shot learning on miniImageNet and reported the results in the table below. We can see that ADV-CE still outperforms CE by **2.33% and 2.24% in 10-shot and 15-shot learning**. This means that ADV-CE does not adversely affect the performance.
> >
> > | Method            |   10-shot    |   15-shot    |
> > |------------------|--------------|--------------|
> > | CE                   | 86.22 ± 0.37 | 87.34 ± 0.35 |
> > | ADV-CE           | 88.55 ± 0.32 | 89.58 ± 0.31 |
> > | Performance Gain | 2.33         | 2.24         |
>
> #### Q9: The main concern is that the core idea is not innovative, similar to the incremental superposition of multiple algorithms.
>
> > - Though VRM itself is not a new algorithm, adapting it and making it work to address the empirical distribution deviation in few-shot learning are novel.
> > - Moreover, we derive the resulting expected vicinal loss functions for both the cross-entropy loss and SVM, which are new and listed as our core contributions.
> > - We also propose to address the issue of how to obtain the adaptive vicinal distribution for each instance for VRM in few-shot learning by harnessing unlabeled data and a lazy random walk algorithm.

---

> ### Author Response · Authors · 2022-08-05
> **Please let us know if you have any further questions.**
>
> Hi Reviewer kfYN,
>
> We would like to follow up to see if our response addresses your concerns or if you have any further questions. We would really appreciate the opportunity to discuss this further if our response has not already addressed your concerns. Thank you again!

---

> ### Comment · Reviewer_kfYN · 2022-08-06
> **Reply**
>
> Thank you for the responses! But the work does not make it above the borderline in my opinion. I will insist on my original comments & rating.
>
> The reasons are as follows:
>
> **(1) Research Problem**
>
> The research problem in this paper has been explored [1]. This contradicts the author's claim to solve an unexplored problem in Lines 32~34.
>
> **(2) Proposed Method**
>
> Although it is novel for the authors to adopt the old VRM algorithm to address the empirical distribution bias in few-shot learning, the whole VRM application process does not bring new challenges. It's hard to convince me that the method of task A applied to task B is a great contribution.
>
> **(3) Experimental Results**
>
> The experimental results of Q6 are also difficult to convince me.
>
> Although ADV shows good performance for the WRN backbones in Table 1, if compared to STOA with different backbones (Table 2), ADV is weaker than STOA in most.
>
> [1] Shuo Yang, Lu Liu, and Min Xu. Free lunch for few-shot learning: Distribution calibration. ICLR, 2021.

---

> > ### Author Response · Authors · 2022-08-07
> > **2nd Response to Reviewer kfYN Part II**
> >
> > **Q13: Experimental Results: if compared to STOA with different backbones (Table 2), ADV is weaker than STOA in most.**
> >
> > > - It is **not true** that "ADV is weaker than STOA in most". We summarize the comparison with SOTA with CONV and ResNet-12 backbones by the results in Table 1 and 2. ADV is better than SOTA baselines in all cases expect the FEAT with CONV in 5 shot.
> > >
> > >   | Method | Backbone | 1 shot | 5 shot |
> > >   | --- | --- | --- | --- |
> > >   | TPN [18] | CONV | 55.51 ± 0.86 | 69.86 ± 0.65 |
> > >   | FEAT [41] | CONV | 57.04 ± 0.20 | **72.89 ± 0.16** |
> > >   | TEAM [22] | CONV | 56.57 | 72.04 |
> > >   | ADV-SVM | CONV | *59.02 ± 0.70* | *72.13 ± 0.53* |
> > >   | ADV-CE | CONV | **59.22 ± 0.68** | 72.07 ± 0.51 |
> > >
> > >   | Method | Backbone | 1 shot | 5 shot |
> > >   | --- | --- | --- | --- |
> > >   | TPN [18] | ResNet | 59.46 | 75.65 |
> > >   | TEAM [22] | ResNet | 60.07 | 75.90 |
> > >   | ADV-SVM | ResNet | *64.27 ± 0.74* | *76.37 ± 0.54* |
> > >   | ADV-CE | ResNet | **65.03 ± 0.74** | **77.12 ± 0.52** |
> > >
> > > - We compared ADV-CE with **3 SOTA methods**: Transductive Information Maximation (TIM) [R1], LaplacianShot [R2], and Obliquer Manifold (OM) [R5].
> > >
> > >   - The performance of few-shot learning is affected by both the base model obtained in the pre-training phase (meta-training phase) and the fine-tuning strategy in the fine-tuning phase (meta-testing phase). **For a fair comparison in the meta-testing phase, the base model should be the same for different methods.**
> > >
> > >   - The reason why we select these 3 methods is that they are all independent of the pre-training of the base model and do not require training any extra module in the pre-training /meta-training phase.
> > >
> > >   - **For a fair comparison**, we feed all methods the same fixed WRN feature extractor and the same 10,000 meta-testing tasks. As TIM and LaplacianShot use the same pre-trained weights of a feature extractor, we use these pre-trained weights for all of the 4 methods. No data augmentation is applied to images in meta-testing tasks.
> > >
> > >   - We present the results in the table below. **We observe that ADV-CE outperforms all baselines in 1-shot learning and obtains the second best performance in 5-shot learning, which is slightly lower than TIM.**
> > >
> > >
> > > | Method | 1-shot | 5-shot |
> > > | --- | --- | --- |
> > > | TIM[R1] | 74.36 ± 0.26 | **84.58 ± 0.15** |
> > > | LaplacianShot[R2] | 69.10 ± 0.23 | 80.86 ± 0.15 |
> > > | OM[R5] | 68.67 ± 0.23 | 83.42 ± 0.15 |
> > > | ADV-CE | **74.92 ± 0.25** | 84.53 ± 0.16 |
> > >
> > > - Note that the major contributions of TIM lie in regularization on query sets, though ours can be deemed as regularization on support sets, **which means its contributions are orthogonal to our contributions**. We are more than interested in integrating TIM to further boost ours. When integrating TIM with ADV-CE and applying it on WRN feature extractor pre-trained by [19] on miniImageNet, ADV-CE+TIM achieves competitive performance to SOTA methods in the following table, where the performances of SOTA methods are reported in the original paper.
> > >
> > >   | Method | 1-shot | 5-shot |
> > >   | --- | --- | --- |
> > >   | TIM [R1] | 77.8 | 87.4 |
> > >   | LaplacianShot [R2] | 74.86 ± 0.19 | 84.13 ± 0.14 |
> > >   | [R3] | 80.04 | 87.64 |
> > >   | [R4] | 76.84 | 84.36 |
> > >   | OM[R5] | 80.64 ± 0.34 | **89.39 ± 0.39** |
> > >   | ADV-CE+TIM | **80.75 ± 0.25** | 88.97 ± 0.14 |
> > >
> > > Note that both [R3] and OM [R5] are integrated with TIM.
> > >
> > > [R1] Boudiaf, Malik, et al. "Information maximization for few-shot learning." NeurIPS 2020.
> > >
> > > [R2] Ziko, Imtiaz, et al. "Laplacian regularized few-shot learning." ICML 2020.
> > >
> > > [R3] Cui, Wentao, and Yuhong Guo. "Parameterless transductive feature re-representation for few-shot learning." ICML 2021.
> > >
> > > [R4] Lee, Dong Hoon, and Sae-Young Chung. "Unsupervised Embedding Adaptation via Early-Stage Feature Reconstruction for Few-Shot Classification." ICML 2021.
> > >
> > > [R5] Qi, Guodong, et al. "Transductive Few-Shot Classification on the Oblique Manifold." ICCV. 2021.

---

> > ### Author Response · Authors · 2022-08-07
> > **2nd Response to Reviewer kfYN Part I**
> >
> > Thanks again for your time and comments. There might be some misunderstanding. We would like to clarify point by point. Please kindly let us know whether you have any further concerns.
> >
> > **Q10: Research Problem.**
> >
> > > - In Lines 32-34, we **do not** claim the research problem is unexplored. Our claim is that the research problem is relatively **"less explored"** than "the majority of meta-learning work revolved around developing better meta-training strategies".
> > >
> > > - We agree that Free-Lunch[1] ([39] in the paper) has explored the research problem of task-specific generalization in meta-testing stage.
> > >
> > >   - In lines 99-105, we acknowledge that our method shared a similar spirit of [1] and summarize the main differences between [1] and ours.
> > >
> > > - Considering the importance of task-specific generalization in meta-testing stage, we believe it merits further exploration.
> > >
> > > - We highlight the advantages of the proposed method over [1].
> > >
> > >   - The vicinal distribution estimated using unlabeled data from the same task in our method is closer to the true distribution than the calibrated distribution estimated in [1].
> > >   - Unlike [1] which optimizes the objective by sampling 750 data instances, our method directly optimizes the expected vicinal loss over the distributions, resulting in more efficient and stable training than [1].
> > >   - Our method significantly outperforms [1] (6% and 3% in 1-shot and 5-shot learning on miniImageNet).
> >
> > **Q11: Proposed Method: the whole VRM application process does not bring new challenges.**
> >
> > > - There are two main challenges as listed below. It may be these challenges that are holding back the popularity of VRM using Gaussian vicinity, even though it has been proposed for more than 20 years.
> > >
> > >   1. It is extremely challenging to derive the expected vicinal loss function over a Gaussian distribution, especially for cross-entropy loss.
> > >
> > >     - We accomplish the challenge by approximating the CE loss using an uncommon technique, namely, a 2nd-order Taylor expansion w.r.t. to data.
> > >
> > >     - To the best of our knowledge, we are the first to derive such vicinal loss for CE and SVM in multi-class classification.
> > >
> > >   2. In few-shot learning, It is difficult to estimate the vicinal distribution for each instance given the limited data.
> > >
> > >     - Our contribution to addressing this challenge is to propose performing lazy random walks among unlabeled data to estimate the vicinal distribution.
> >
> > **Q12: The experimental results of Q6 are also difficult to convince me.**
> >
> > >
> > > - We analyze why the performance of 5-step ADV-CE is competitive with 100-step ADV-CE.
> > >
> > >   - We use a good initialization and a large learning rate for a fast convergence of the 5-step ADV-CE.
> > >
> > >     - The initialization is the prototype of each class.
> > >
> > >     - The learning rate is 10 times that of 100-step ADV-CE.
> > >
> > >   - Since the learning rate may be too large, or the number of step too small, the 5-step ADV-CE converges to a sub-optimal solution, achieving a lower performance than 100-step ADV-CE.
> > >
> > > - If using the same learning rate as 100-step ADV-CE, the performance of 5-step ADV-CE is 71.81 and 83.90 for 1-shot and 5-shot learning.

---

> > > ### Comment · Reviewer_kfYN · 2022-08-10
> > > **Reply**
> > >
> > > I still think that the method is not novel and the experimental part is too weak. This is not above the borderline in my opinion.

---

> > > > ### Author Response · Authors · 2022-08-10
> > > > **Thank Reviewer kfYN and a follow-up regarding the experimental part**
> > > >
> > > > First, we again post part II of the previous response here for your reference, in case that you have missed that part which is in the other post.
> > > >
> > > > Second, if the following response still fails to address your concerns, we would very much appreciate if you elaborate the weakness of our experiments.
> > > >
> > > > **Q13: Experimental Results: if compared to STOA with different backbones (Table 2), ADV is weaker than STOA in most.**
> > > >
> > > > > - It is **not true** that "ADV is weaker than STOA in most". We summarize the comparison with SOTA with CONV and ResNet-12 backbones by the results in Table 1 and 2. ADV is better than SOTA baselines in all cases expect the FEAT with CONV in 5 shot.
> > > > >
> > > > >   | Method | Backbone | 1 shot | 5 shot |
> > > > >   | --- | --- | --- | --- |
> > > > >   | TPN [18] | CONV | 55.51 ± 0.86 | 69.86 ± 0.65 |
> > > > >   | FEAT [41] | CONV | 57.04 ± 0.20 | **72.89 ± 0.16** |
> > > > >   | TEAM [22] | CONV | 56.57 | 72.04 |
> > > > >   | ADV-SVM | CONV | *59.02 ± 0.70* | *72.13 ± 0.53* |
> > > > >   | ADV-CE | CONV | **59.22 ± 0.68** | 72.07 ± 0.51 |
> > > > >
> > > > >   | Method | Backbone | 1 shot | 5 shot |
> > > > >   | --- | --- | --- | --- |
> > > > >   | TPN [18] | ResNet | 59.46 | 75.65 |
> > > > >   | TEAM [22] | ResNet | 60.07 | 75.90 |
> > > > >   | ADV-SVM | ResNet | *64.27 ± 0.74* | *76.37 ± 0.54* |
> > > > >   | ADV-CE | ResNet | **65.03 ± 0.74** | **77.12 ± 0.52** |
> > > > >
> > > > > - We compared ADV-CE with **3 SOTA methods**: Transductive Information Maximation (TIM) [R1], LaplacianShot [R2], and Obliquer Manifold (OM) [R5].
> > > > >
> > > > >   - The performance of few-shot learning is affected by both the base model obtained in the pre-training phase (meta-training phase) and the fine-tuning strategy in the fine-tuning phase (meta-testing phase). **For a fair comparison in the meta-testing phase, the base model should be the same for different methods.**
> > > > >
> > > > >   - The reason why we select these 3 methods is that they are all independent of the pre-training of the base model and do not require training any extra module in the pre-training /meta-training phase.
> > > > >
> > > > >   - **For a fair comparison**, we feed all methods the same fixed WRN feature extractor and the same 10,000 meta-testing tasks. As TIM and LaplacianShot use the same pre-trained weights of a feature extractor, we use these pre-trained weights for all of the 4 methods. No data augmentation is applied to images in meta-testing tasks.
> > > > >
> > > > >   - We present the results in the table below. **We observe that ADV-CE outperforms all baselines in 1-shot learning and obtains the second best performance in 5-shot learning, which is slightly lower than TIM.**
> > > > >
> > > > >
> > > > > | Method | 1-shot | 5-shot |
> > > > > | --- | --- | --- |
> > > > > | TIM[R1] | 74.36 ± 0.26 | **84.58 ± 0.15** |
> > > > > | LaplacianShot[R2] | 69.10 ± 0.23 | 80.86 ± 0.15 |
> > > > > | OM[R5] | 68.67 ± 0.23 | 83.42 ± 0.15 |
> > > > > | ADV-CE | **74.92 ± 0.25** | 84.53 ± 0.16 |
> > > > >
> > > > > - Note that the major contributions of TIM lie in regularization on query sets, though ours can be deemed as regularization on support sets, **which means its contributions are orthogonal to our contributions**. We are more than interested in integrating TIM to further boost ours. When integrating TIM with ADV-CE and applying it on WRN feature extractor pre-trained by [19] on miniImageNet, ADV-CE+TIM achieves competitive performance to SOTA methods in the following table, where the performances of SOTA methods are reported in the original paper.
> > > > >
> > > > >   | Method | 1-shot | 5-shot |
> > > > >   | --- | --- | --- |
> > > > >   | TIM [R1] | 77.8 | 87.4 |
> > > > >   | LaplacianShot [R2] | 74.86 ± 0.19 | 84.13 ± 0.14 |
> > > > >   | [R3] | 80.04 | 87.64 |
> > > > >   | [R4] | 76.84 | 84.36 |
> > > > >   | OM[R5] | 80.64 ± 0.34 | **89.39 ± 0.39** |
> > > > >   | ADV-CE+TIM | **80.75 ± 0.25** | 88.97 ± 0.14 |
> > > > >
> > > > > Note that both [R3] and OM [R5] are integrated with TIM.
> > > > >
> > > > > [R1] Boudiaf, Malik, et al. "Information maximization for few-shot learning." NeurIPS 2020.
> > > > >
> > > > > [R2] Ziko, Imtiaz, et al. "Laplacian regularized few-shot learning." ICML 2020.
> > > > >
> > > > > [R3] Cui, Wentao, and Yuhong Guo. "Parameterless transductive feature re-representation for few-shot learning." ICML 2021.
> > > > >
> > > > > [R4] Lee, Dong Hoon, and Sae-Young Chung. "Unsupervised Embedding Adaptation via Early-Stage Feature Reconstruction for Few-Shot Classification." ICML 2021.
> > > > >
> > > > > [R5] Qi, Guodong, et al. "Transductive Few-Shot Classification on the Oblique Manifold." ICCV. 2021.

---

### Official Review · Reviewer_b8EE · 2022-07-13

**Rating:** 7
**Confidence:** 3
**Soundness:** 3 good
**Presentation:** 3 good
**Contribution:** 3 good

**Summary:**

The goal of the paper is to improve the robustness of classifier for task-specific generalisation in the meta-testing stage.  The work introduces Vicinal Risk Minimization in which a Gaussian distribution is fitted on the unlabeled data samples and the probabilities of the labeled samples are obtained using lazy random walk. The work proved the expected vicinal loss functions for cross-entropy loss and SVM. Experiments on miniImageNet, CUB, CIFAR-FS for few-shot learning demonstrate the effectiveness of the proposed approach.

**Questions:**

* "Our approach also considers Gaussian vicinities but estimates the mean by the unlabeled data, which is generally not on the training samples". How is this a better formulation?

* What is the dimensionality of the Gaussians being considered here? How realistic is the approach as the dataset scales both in terms of number of samples and size of each image.


**Limitations:**

The limitations have been adequately addressed.

**Strengths And Weaknesses:**

Strengths and Weaknesses:

* The paper is well-motivated and easy to follow. The overall idea of constructing vicinal distribution of the training samples using unlabeled training data is novel

* The experimental set-up is sound and demonstrates the usefulness of the approach. The results show the improvement in the generalization performance of the models with the proposed method. The experiments are extensive and outline the set-up clearly.

* Thorough ablations in Table 3 clearly show the advantage of using the VRM principle over the CE-Mixup principle.



There are no major weaknesses in the paper, however, certain claims in the manuscript can be supported with citations. There are a few typos and can be corrected in the revision.



 - Minor line 36: “the significant deviation of the empirical distribution..”  Add citation here?

 - Line 81: benchmars-> benchmarks

 - Line 99: It [37] -> In

 - Lime 110: . Besides, (author?)

 - Line 126: tge -> the

 - Line 133: expect loss -> expected loss

 - Line 177 it replace->replaces

---

> ### Author Response · Authors · 2022-08-02
> **Response to Reviewer b8EE**
>
> We sincerely thank you for highlighting the strengths of this work. We detail our response to your concerns below point by point. Please kindly let us know if our response addresses the issues you raised in this paper.
>
> #### Q1: How is this a better formulation: the vicinal distribution obtained by lazy random walk vs. the vicinal distribution located in the original data
>
> > - The empirical distribution (estimated from the original data) may deviate from the true distribution. For example, we illustrate in Fig 1(a)(b) and Fig 2 that the prototypes in 1-shot learning may deviate from the true class mean. Therefore, the vicinal distribution whose mean is located in the original instance is not optimal and cannot significantly improve the classifier. This motivates us to choose a better vicinal distribution that is closer to the true distribution. We obtain such adaptive vicinal distributions by performing a lazy random walk in unlabeled data for each support data.
>
> > - We performed an experiment to compare the performance of i) **ADV-CE** using the adaptive vicinal distribution (both mean and variance are estimated by lazy random walk); ii) **ADV-CE-VAR**, using vicinal distribution located in the original data (only the variance is estimated by a lazy random walk. It mean remains the same as original feature.) and iii) **CE** without using vicinal distribution. Their performances on miniImageNet are:
> >
> > | Methods    | 1-Shot       | 5-shot       |
> > |------------|--------------|--------------|
> > | ADV-CE     | 74.63 ± 0.48 | 86.12 ± 0.29 |
> > | ADV-CE-VAR | 66.47 ± 0.62 | 84.51 ± 0.41 |
> > | CE         | 65.38 ± 0.52 | 83.33 ± 0.30 |
> >
> > - From the results, we can see that ADV-CE significantly outperforms ADV-CE-VAR. ADV-CE-VAR gains a small improvement over CE from the VRM. The results verifies that the vicinal distribution obtained by lazy random walk is much better than the vicinal distribution whose mean is located in the original data.
>
> #### Q2: What is the dimensionality of the Gaussians? How realistic is the approach as the dataset scales both in terms of number of samples and size of each image.
>
> > - The vicinal distribution of the distribution of the vicinity or neighborhood around a data point in the feature space. The dimension of the vicinal distribution is the same as the feature dimension and is independent of the number of samples and the input size of each image. So there is no scaling problem.

---

> > ### Comment · Reviewer_b8EE · 2022-08-09
> > **Response**
> >
> > The rebuttal has addressed most of my concerns. I keep my rating and recommend acceptance.

---

### Author Response · Authors · 2022-08-07
**About Paper Revision**

Dear Reviewers,

We have submitted a revision of the paper. The main changes are summarized below.

- We corrected typos and grammatical errors.

- We revised sections 3.2.2 and 3.3 to give a clearer presentation of Vicinal SVM and the lazy random walk algorithm for obtaining the adaptive vicinal distribution.

- We added additional experimental results on the comparison with SOTA methods in the appendix.

The main changes are highlighted in blue.

---

### Meta-Review · Area_Chair_SXww · 2022-08-30

**Recommendation:** Accept
**Confidence:** Certain

**Metareview:**

In this paper, authors study a few-shot learning setting where the training distribution deviates from true distribution. To achieve a more accurate approximation of the true distribution, authors propose assuming Gaussian-like vicinal distribution around each training data point which results in a vicinal loss. Extensive empirical results in the paper show that the proposed method improves over the baseline methods. While vicinal loss has been used in other settings, this problem formulation and use of vicinal loss is novel and the empirical advantage is significant. Therefore, I am recommending acceptance. Given reviewers' concerns about typos and presentation issues, I encourage the authors to make a few more passes over the paper and improve the writing and presentation.

**Award:**

No

---

### Decision · Program_Chairs · 2022-09-14

Accept